# Narrative event segmentation in the cortical reservoir

**Peter Ford Dominey**[1,2]*

1 INSERM UMR1093-CAPS, Université Bourgogne Franche-Comté, UFR des Sciences du Sport, Dijon,
2 Robot Cognition Laboratory, Institute Marey, Dijon

* peter.dominey@inserm.fr

## Abstract

Recent research has revealed that during continuous perception of movies or stories, humans display cortical activity patterns that reveal hierarchical segmentation of event structure. Thus, sensory areas like auditory cortex display high frequency segmentation related to the stimulus, while semantic areas like posterior middle cortex display a lower frequency segmentation related to transitions between events. These hierarchical levels of segmentation are associated with different time constants for processing. Likewise, when two groups of participants heard the same sentence in a narrative, preceded by different contexts, neural responses for the groups were initially different and then gradually aligned. The time constant for alignment followed the segmentation hierarchy: sensory cortices aligned most quickly, followed by mid-level regions, while some higher-order cortical regions took more than 10 seconds to align. These hierarchical segmentation phenomena can be considered in the context of processing related to comprehension. In a recently described model of discourse comprehension word meanings are modeled by a language model pre-trained on a billion word corpus. During discourse comprehension, word meanings are continuously integrated in a recurrent cortical network. The model demonstrates novel discourse and inference processing, in part because of two fundamental characteristics: real-world event semantics are represented in the word embeddings, and these are integrated in a reservoir network which has an inherent gradient of functional time constants due to the recurrent connections. Here we demonstrate how this model displays hierarchical narrative event segmentation properties beyond the embeddings alone, or their linear integration. The reservoir produces activation patterns that are segmented by a hidden Markov model (HMM) in a manner that is comparable to that of humans. Context construction displays a continuum of time constants across reservoir neuron subsets, while context forgetting has a fixed time constant across these subsets. Importantly, virtual areas formed by subgroups of reservoir neurons with faster time constants segmented with shorter events, while those with longer time constants preferred longer events. This neuro-computational recurrent neural network simulates narrative event processing as revealed by the fMRI event segmentation algorithm provides a novel explanation of the asymmetry in narrative forgetting and construction. The model extends the characterization of online integration processes in discourse to more extended narrative, and demonstrates how reservoir computing provides a useful model of cortical processing of narrative structure.

**Data Availability Statement:** https://github.com/pfdominey/Narrative-Integration-Reservoir/.

**Funding:** PFD received funding from the French Région Bourgogne Franche Comté, Grant ANER RobotSelf 2019-Y-10650. The funders had no role in study design, data collection and analysis,

decision to publish, or preparation of the
manuscript.

**Competing interests:** The authors have declared
that no competing interests exist.

## Author summary

When we watch movies or listen to stories, our brains are led through a trajectory of activation whose structure reflects that of the event structure of the story. This takes place at multiple timescales across the brain, likely corresponding to different timescales of event representation. While this has been well described in human fMRI, the underlying computations that lead to these activation trajectories has not yet been fully characterized. The current research develops and explores a recurrent network "reservoir" model of cortical computation, whose natural internal dynamics help to provide an explanation of the trajectory of brain states that are observed in different cortical areas in humans. The model is exposed to narratives in the form of word embeddings for words in the narrative transcript. Neural activation in the model reveals event structure at multiple levels of temporal structure. This begins to provide insight into the computations underlying the event structure observed in the human brain during narrative processing.

## Introduction

Human existence is embedded in a never ending flow of time [1]. A major function of the nervous system is to segment and organize the spatiotemporal flow of perception and action into coherent and relevant structure for effective behavior [2–4]. A particular challenge is to take into account the context of the recent and distant past while addressing the current situation. One of the highest expressions of this is human narrative, which provides a mechanism for encoding, transmitting and inventing reality with its inherent temporal structure [5–7]. Recent advances in neuroscience recording and analysis methods have made possible the study of temporal structure of neurophysiological signals in the human brain during narrative processing, e.g. [8–10].

One of the questions addressed in such research concerns the actual computations that integrate past and present information within the hierarchical networks in the brain [9]. Interestingly, networks of neurons that have recurrent connections are particularly well suited for problems that require integration of the past with the present [11–14]. When we ask how such recurrent networks might be implemented in the brain it is even more interesting that one of the principal characteristics of cortical connectivity is the high density of local recurrent connections [15], i.e. that one of the primary characteristics of primate cortex is that it is a recurrent network.

It is thus not surprising that recurrent networks have been used to model cortical processing of sequential and temporal structure, explaining both behavior and neurophysiology [16–20]. A subset of recurrent network models use fixed (rather than modifiable) random connections which avoids truncating the recurrent dynamics as required by recurrent learning mechanisms [11]. This allows the full expression of recurrent dynamics and high dimension expression of the inputs in the recurrent dynamics. This reservoir computing framework has been used to explain aspects of primate cortical neurophysiology in complex cognitive tasks [18,19,21,22].

Reservoir computing refers to a class of neural network models that exploit the rich dynamics that are created when information circulates in recurrent connections within the network [14,23–26]. The novelty is that instead of employing learning-related modifications of weights within the recurrent network, the recurrent connections are fixed, and learning occurs only in connections between the units in the recurrent network and the so-called readout neurons.

This eliminates the truncation of the recurrent activations and other simplifications that are necessary in the implementation of recurrent learning algorithms [27], and allows a natural way to use temporal structure in the input signal. In one of the first implementations of reservoir computing, Dominey et al. [19] proposed that the primate frontal cortex could be modeled as a reservoir, and the striatum as the readout, where corticostriatal readout connections are modified by reward-related dopamine. They used this model to simulate primate sensorimotor sequence learning, and the corresponding neural activity in prefrontal cortex. Again, the motivation for using fixed recurrent connections is that learning algorithms for recurrent connections require a simplification or truncation of the recurrent network dynamics. With fixed recurrent connections, the reservoir displays a rich, high dimensional mixture of its inputs, and a rich temporal structure. The model thus learned the sensorimotor sequencing task, and the reservoir units displayed a high dimensional mixture of space and time as observed in the primate prefrontal cortex [28]. Indeed, the somewhat surprising phenomenon underlying reservoir computing is that without learning, the random recurrent connections produce a high dimensional representation that encodes an infinity of nonlinear re-combinations of the inputs over time, thus providing universal computational power [23]. The appropriate representation in the high dimensional representation can then be retrieved through learning with a simple linear readout.

Reservoir units are leaky integrators that have their own inherent time constant. The interaction of such units via recurrent connections generates a further diversity of effective time constants within the system [29]. Thus, reservoirs have an inherent capacity to represent time and spatiotemporal structure [16,30–34]. That is, the recurrent connections produce a form of fading reverberation of prior inputs so that the reservoir is inherently capable of representing temporal structure.

In typical use, an input signal excites the reservoir, generating a spatiotemporal trajectory of activation, and the connections to the readout neurons are trained so as to produce the desired behavior. In Enel et al. [18], the inputs were a sequence of activation of spatial targets simulating an explore-exploit task where a monkey first explored to identify which of four targets was rewarded, and then exploited or repeated choices of the rewarded target until a new exploration began. Readout neurons were trained so as to produce the correct exploration or exploitation responses on each trial. Interestingly, the units in the recurrent network displayed a nonlinear mixture of the input signal, referred to as mixed selectivity [22], which was highly correlated with the neural responses recorded in the behaving monkey. Thus, reservoir models of cortical function yield interesting observations at two levels: First, in terms of learned behavioral output, and second, in terms of the high dimensional representations within the reservoir, independent of learning.

Reservoir models have been used to perform semantic role labeling in sentence processing [35]. In the domain of narrative and discourse comprehension, Uchida et al. [36] recently used a reservoir-based model to explain how discourse context is integrated on-line so that each new incoming word is processed in the integrated context of the prior discourse. The model addressed the immediacy constraint, whereby all past input is always immediately available for ongoing processing [37,38]. To address this constraint, the model proposed that making past experience immediately accessible in discourse or narrative comprehension involves a form of temporal-to-spatial integration on two timescales. The first timescale involves the integration of word meaning over extended lifetime, and corresponds to the notion of lexical semantics. This is modeled by the Wikipedia2Vec language model [39]. Wikipedia2vec uses three models to generate word embeddings: a word2vec skip-gram model [40] applied to the 3 billion word 2018 Wikipedia corpus, the link-graph semantics extracted from Wikipedia, and anchor-context information from hyperlinks in Wikipedia [39]. The second timescale is at the level of the

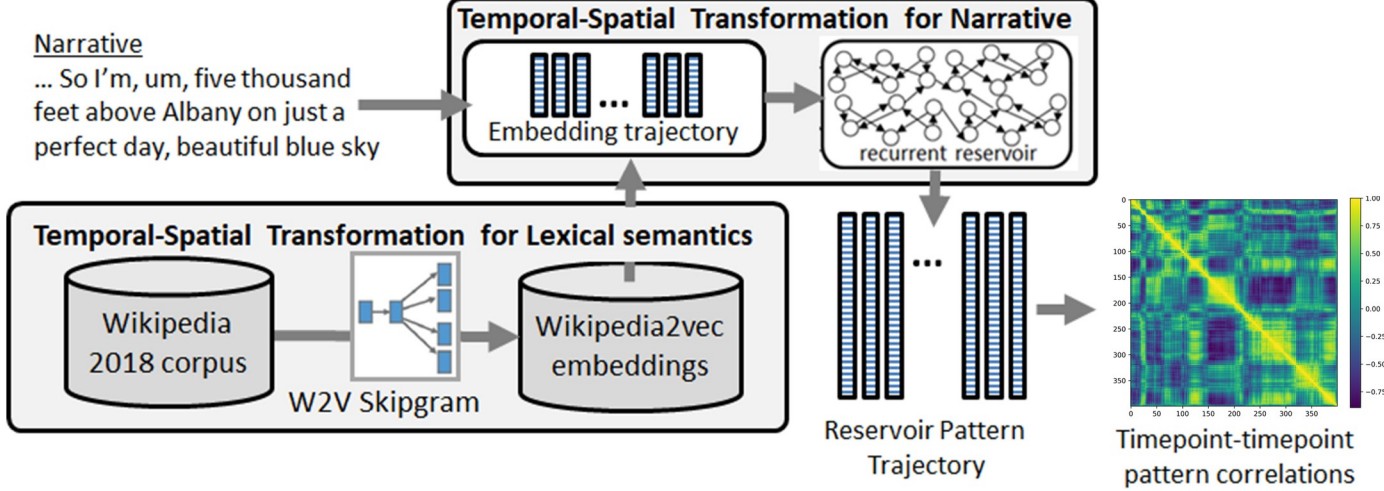

**Fig 1. Narrative Integration Reservoir.** Word embeddings, generated by Wikipedia2vec, input to reservoir which generates a trajectory of internal states that represent the word-by-word processing of the narrative.

on-line integration of words in the narrative. This is modeled by a recurrent reservoir network that takes as input the sequence of word embeddings corresponding to the successive words in the discourse. The model is illustrated in Fig 1. In [36], the reservoir was trained to generate the discourse vector, a vector average of the input words in the discourse. A sequence of word embeddings was input to the reservoir, and the readouts were trained to generate the average of the input embeddings.

The model was used to simulate human brain responses during discourse comprehension as the N400 ERP, a neurophysiological index of semantic integration difficulty [41]. N400 amplitude increases as a function of the semantic distance between a target word and the prior discourse [42,43]. This neurocomputational model was the first to simulate immediacy and overrule in discourse-modulated N400. It is important to note that this research exploited reservoir computing in terms of the behavior of the trained model, as observed in the trained readout neurons, with respect to human behavior and N400 responses. This research did not examine the coding within the recurrent reservoir itself.

The current research explores how this model can account for human neurophysiology of narrative processing in much more extended narratives, increasing from the order of $10^1$ to $10^2$–$10^3$ words. Here, instead of looking at trained readout responses, we directly examine the high dimensional representations of word embeddings within the recurrent reservoir, which is a high dimensional non-linear integrator. We compare these representations with those from human neurophysiology in human fMRI. It is important to acknowledge that this research would not be possible without the open science approach in modern neuroscience and machine learning. Data and algorithms used in the analysis of the human neuroscience are publicly available [8,44,45], as is python code for developing the reservoir model [46], and for creating word embeddings for narrative transcripts [39]. This open policy creates a context where it is possible to perform the same analyses on human fMRI and on Narrative Integration Reservoir simulations.

The Narrative Integration Reservoir model embodies two hypotheses: First, that word embeddings from a sufficiently large corpus can serve as a proxy for distributed neural semantics. It is important to note that these embeddings contain extensive knowledge of events as we will see below. Second, that temporal-spatial integration of these word vectors in a recurrent

reservoir network simulates cortical temporal-spatial integration of narrative. To test these hypotheses the model is confronted with two principal observations related to cortical processing of narrative. The first has to do with the appearance of transitions between coherent epochs of distributed cortical activation corresponding to narrative event boundary segmentation [8]. The second has to do with a hierarchy of time constants in this processing, and an asymmetry in time constants for constructing vs. forgetting narrative context [9].

Baldassano et al. [8] developed a novel algorithm to detect narrative event boundaries in the fMRI signal of subjects listening to narratives. Their algorithm is a variant of the hidden Markov model (HMM). It identifies events as continuous sequences in the fMRI signal that demonstrate high similarity with an event-specific pattern, and event transitions as discontinuities in these patterns. They used the HMM to segment fMRI activity during narrative processing and made several remarkable observations. In particular they demonstrated that segmentation granularity varies along a hierarchy from short events in sensory regions to long events in high order areas (e.g. TPJ) representing abstract, multimodal situation models.

This allows us to pose the question, are these higher level representations in areas like TPJ the result of longer effective time constants in these higher cortical areas, or is there some processing taking place that is imposing these longer time constants on these higher processing areas? Indeed, in a large distributed recurrent network it is likely that different populations of neurons with different effective time constants will emerge, as demonstrated by Bernacchia et al. [29]. We can thus predict that a distribution of neurons with different time constants will be observed in the reservoir and that these time constants will be related to aspects of their narrative segmentation processing.

Interestingly Chien and Honey [9] demonstrated that the expression of such time constants in narrative processing is dependent on context. In their experiment, one group of subjects listened to an intact narrative (e.g. with a structure ABCD), and another to a narrative that had been scrambled (e.g. with a structure ACBD). The narrative was "It's Not the Fall that Gets You" by Andy Christie (https://themoth.org/stories/its-not-the-fall-that-gets-you). The fMRI responses were then compared across these groups in two contexts. The *forgetting* context compared the situation where the two groups started by hearing the same narrative, and then shifted to two different narratives (e.g. AB in group 1 and AC in group 2). Thus the forgetting context was the transition from same (A) to different (B/C). The *construction* context compared the situation where the two groups started hearing different narrative and then began to hear the same narrative (e.g. CD in group 1 and BD in group 2). The transition from different (C/B) to same (D) was the construction context. In this clever manipulation of forgetting and constructing context, Chien and Honey [9] discovered that in different cortical areas the time constants of event structure such as observed by Baldassano et al. [8] is reflected in the rate of context construction, whereas there no systematic relation with context forgetting across these cortical areas. That is, higher cortical areas (e.g. TPJ) construct contexts more slowly, whereas forgetting did not increase in this systematic manner. In order to account for these results, they developed a hierarchical auto-encoder in time (HAT) model. Each hierarchical level in the model receives prior context from the level below, and higher levels have longer explicit time constants, so their context is less influenced by their input at each time step. This allows the system to account for the hierarchy of timescales in context construction. A surprise signal (that is triggered at event segment boundaries) gates the integration of new input and thus allows for a uniform and rapid reset in the case of context forgetting. These observations on event segmentation, the relation between timescales of processing and segmentation granularity, and the asymmetry in timescales for context forgetting and construction provide a rich framework and set of observations against which we can compare the Narrative Integration Reservoir.

In our experiments, the input to the reservoir of the word vectors for each successive word in the narrative produces a spatiotemporal trajectory of reservoir states. This reservoir activation trajectory simulates the fMRI signal of humans that listen to the same narrative, and we can thus apply the segmentation HMM of Baldassano et al. [8] to these reservoir state trajectories.

To evaluate the Narrative Integration Reservoir, we first expose it to a set of short texts extracted from the New York Times, and Wikipedia to test whether the reservoir will generate activity patterns that can be segmented by the HMM. At the same time, we also examine how the HMM can segment the unprocessed embeddings themselves, as well as a linear integration of the embeddings, in order to see what information is available independent of the reservoir.

Next we compare reservoir and human brain neural activity trajectories generated by exposure to the same narrative. We then undertake two more significant tests of the model. First, we examine whether the model can provide insight into a form of asymmetry of construction vs forgetting of narrative context as observed by Chien and Honey [9]. We then test the hypothesis that different effective time constants for reservoir neurons in context construction [9] will be associated with different granularity of event segmentation as observed in fMRI data by Baldassano et al. [8].

## Results

### Segmentation at topic boundaries

We first tested the hypothesis that the Narrative Integration Reservoir when driven by narrative input should exhibit event-structure activity with HMM segmentation correlated with known narrative boundaries. We created a test narrative with clearly identifiable ground truth boundaries. We choose four short text segments from different articles in the New York Times along with four short text segments from 4 Wikipedia articles and concatenated these together to form a single narrative text that had well defined ground truth topic boundaries. This yielded our test narrative which was then used as input to the Narrative Integration Reservoir in order to generate the trajectory of reservoir activation. We tested the resulting event-structured activity by applying the HMM segmentation model of Baldassano et al. [8] to the trajectory of reservoir states generated by feeding the embeddings for the words in these texts into the reservoir. The HMM takes as input the neural trajectory to be segmented, and the number of segments, k, to be identified, and produces a specification of the most probable event boundaries. The HMM model is available in the python BrainIAK library described in the Materials and Methods section.

Independent of the integration provided by the reservoir, there is abundant rich information in the embeddings themselves. It is thus of interest to determine if this information alone is sufficient to allow good segmentation based on the embeddings alone. Likewise, it is crucial to demonstrate that the NIR model's performance is different from something very simple like feeding the embeddings into a linear integrator (LI) model. We thus investigated how the HMM would segment the embeddings, the LI (see Materials and Methods section) with different time constants, and the NIR reservoir with three different leak rates. The results are illustrated in Fig 2.

Each panel in Fig 2 illustrates the HMM segmentation superimposed on the timepoint-timepoint correlations for the different signals: the raw embeddings, the LI with three values for leak rate $\alpha$ (.2, .1, .05), and the NIR reservoir with three leak rates (.2, .1, .05). Starting at $k = 8$ we run the HMM and increase k until the 8 segments are identified. This yields $k = 10$. The dotted lines indicate event boundaries identified by the HMM with $k = 10$, and dots indicate the actual ground truth boundaries.

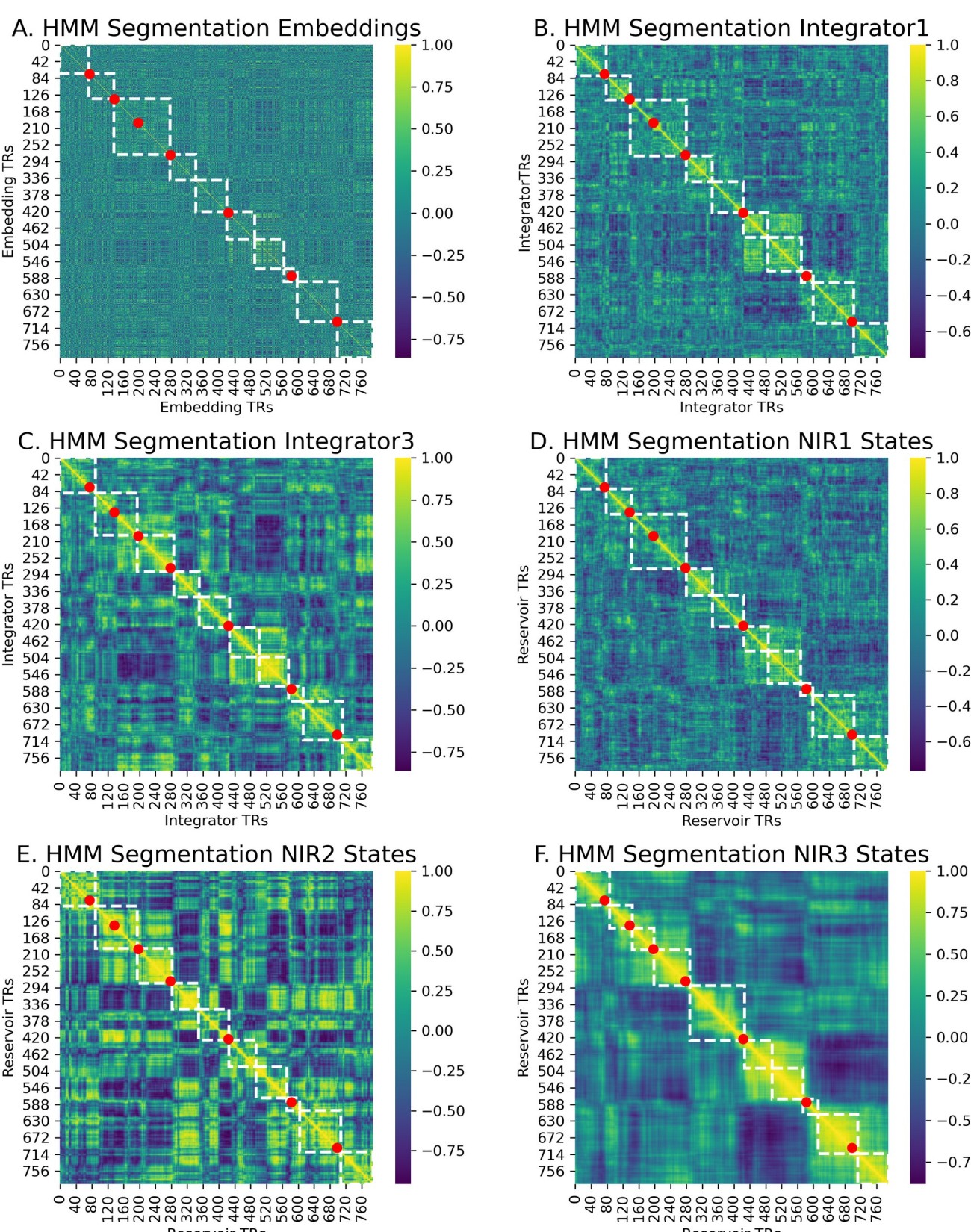

**Fig 2.** HMM segmentation illustrated on time-point time-point correlations for (A) embeddings alone, linear integrator of embeddings with (B) fast leak rate, and (C) slower leak rate, and reservoir with three progressively slower leak rates (D-F). Input is a text made up of 8 paragraphs extracted from Wikipedia (4) and New York Times (4) articles. White dotted lines indicate segmentation of the HMM. Red dots indicate ground truth section boundaries in the text. Note that already the embeddings and linear integrator contain structured information related to the text.

Panel A illustrates segmentation on the raw embeddings. We can see that there is some structure related to the ground truth. Panels B and C illustrate segmentation with LI with values of the leak rate α = 0.2 and 0.05. As α decreases, the memory of past embeddings increases, and the impact of the current input decreases. We observe that the linear integrator is able to segment the narrative. Panels C-E illustrate segmentation on the NIR model reservoir states with the three different leak rates α = 0.2, 0.1 and 0.05. Again, as α increases, the memory of past events increases. We can observe that the NIR representations allow segmentation that is aligned with the ground truth, particularly for α = 0.05.

To evaluate segmentation of the embeddings, the LI, and NIR models, we applied the randomization procedure from Baldassano et al. [8]. For each method, we generate 1000 permuted versions of the HMM boundaries in which the distribution of event lengths (the distances between the boundaries) is held constant but the order of the event lengths is shuffled. We use this null model for comparison with how often the HMM boundaries for the ground truth and those for the model boundaries will be within 3 TRs of each other by chance. The true match value was compared to this distribution to compute a z value, which was converted to a p value. Indeed, all of the methods (embeddings alone, 3 linear integrators, 3 reservoirs) yielded representations whose segmentations correspond to the ground truth and differed significantly from the null model. With k = 8, we observed the following p values: embeddings alone 4.42e-04, LI1 7.21e-09, LI2 9.56e-05, LI3 3.56e-02, NIR1 8.13e-05, NIR2 2.67e-02, NIR3 4.67e-02. With k = 10 all values remained significant except for LI2 and LI3. As illustrated in panel C, LI3 segmentation produces more uniform segments that do not differ significantly from the null model, but align well with the ground truth.

This indicates that already in the embeddings there is sufficient information to segment text from different sources, and that this information can be represented in a linear integration of the embeddings. We can now investigate the possible added value of the non-linear integration provided by the reservoir.

## Comparison of narrative integration reservoir and human fMRI segmentation on corresponding input

Baldassano et al. [8] demonstrated cross-modal segmentation, where application of their HMM to fMRI from separate subject groups that either watched a 24 minute movie or listened to an 18 minute audio narration describing events that occurred in the movie produced similar event segmentation using their HMM. The HMM thus revealed significant correspondences between event segmentation of fMRI activity for movie-watching subjects and audio-narration subjects exposed to the same story with different modalities (movie vs. audio) and different timing. This extended the related work of Zadbood et al. [47] who showed that event-specific neural patterns observed as participants watched the movie were significantly correlated with neural patterns of naïve listeners who listened to the spoken description of the movie.

Using material from these experiments, we set out to compare HMM segmentation in two conditions: The first is the fMRI signal from humans who watched the same 24 minute movie, and the second is the recurrent reservoir activity when the Narrative Integration Reservoir is exposed to the text transcript of the 18 minute recall of that same movie as in [47]. Thanks to their open data policy [8,9,44,45,47], we have access to fMRI from human subjects who

watched the episode of Sherlock, along with a transcript of the recall of this movie. For fMRI we use data from Chen et al. 2017 that was recorded while subjects watched the first 24 minutes of the Sherlock episode (see Materials and Methods). For input to the NIR we use the transcript of the 18 minute audio recall of this 24 minute film from episode from [47]. We use the transcript to generate a sequence of word embeddings from Wikipedia2vec which is the input to the Narrative Integration Reservoir. The resulting trajectory of activation patterns in the reservoir can be segmented using the HMM. In parallel the HMM is used to segment human fMRI from subjects who watched the corresponding movie. This allows a parallel comparison of the HMM segmentation of human brain activity and of Narrative Integration Reservoir activity, as illustrated in Fig 3.

fMRI Data from 16 subjects who watched the 24 minute move were compared with state trajectories from 16 NIR reservoirs exposed to the transcript from the 18 recall of the movie. The movie duration of 24 minutes corresponds to a total recording of 946 TRs. For each subject, the HMM was run on the fMRI data from the angular gyrus (AG) which has been identified as an area that produces related event processing [45,47] and segmentation [8] for watching a movie and listening to recall of the same movie. Baldassano et al. [8] determined that the optimal segmentation granularity for AG is between 50–90 segments for the 1976 TRs in the 50 minute episode, corresponding to a range of 24–43 segments for the 946 TRs in the 24 minute fMRI data we used. We thus chose k = 40 for the HMM segmentation, as a value that has been established to be in the optimal range for AG. The summary results of the HMM segmentation of the NIR and fMRI trajectories are presented in Fig 4. In Fig 4A we see the segmentation into 40 events in the average over all subjects.

For the Narrative Integration Reservoir, 16 instances were created using different seed values for initialization of the reservoir connections, and each was exposed to the word by word narrative of the recall of the movie. For each word, the corresponding Wikipedia2vec 100

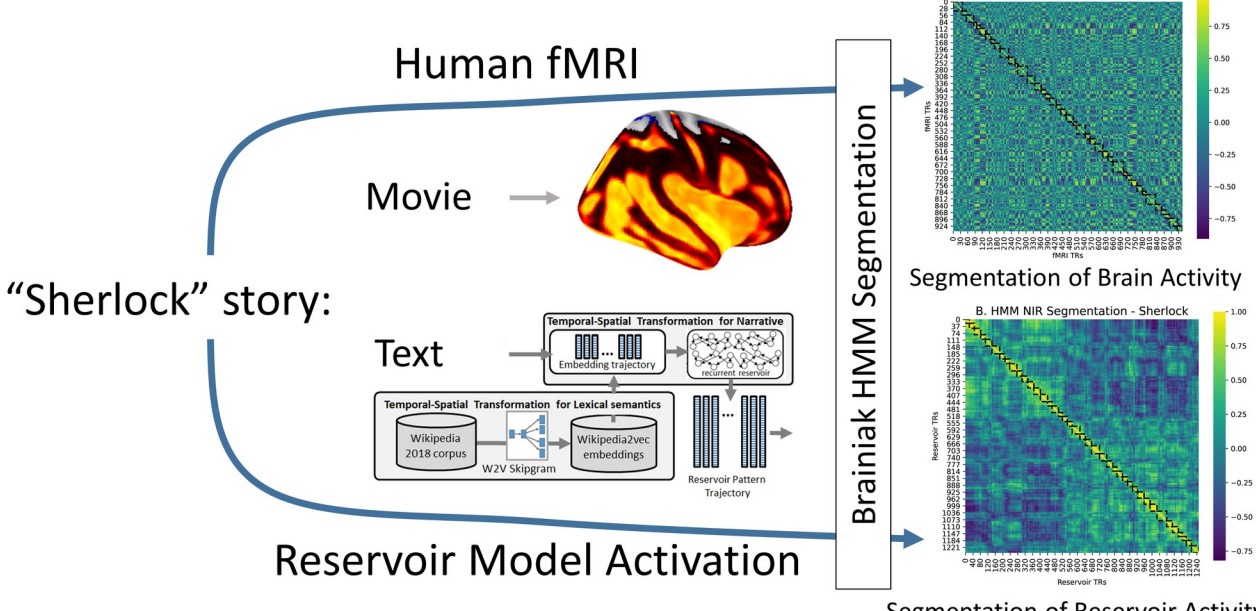

**Fig 3. Pipeline for comparing HMM segmentation of human fMRI and reservoir recurrent states during processing of the same narrative.** Humans watch a "Sherlock" episode in fMRI scanner. Narrative Integration Reservoir model exposed to recall transcript of the same episode. Resulting trajectories of human fMRI and model reservoir states are processed by the Baldassano HMM.

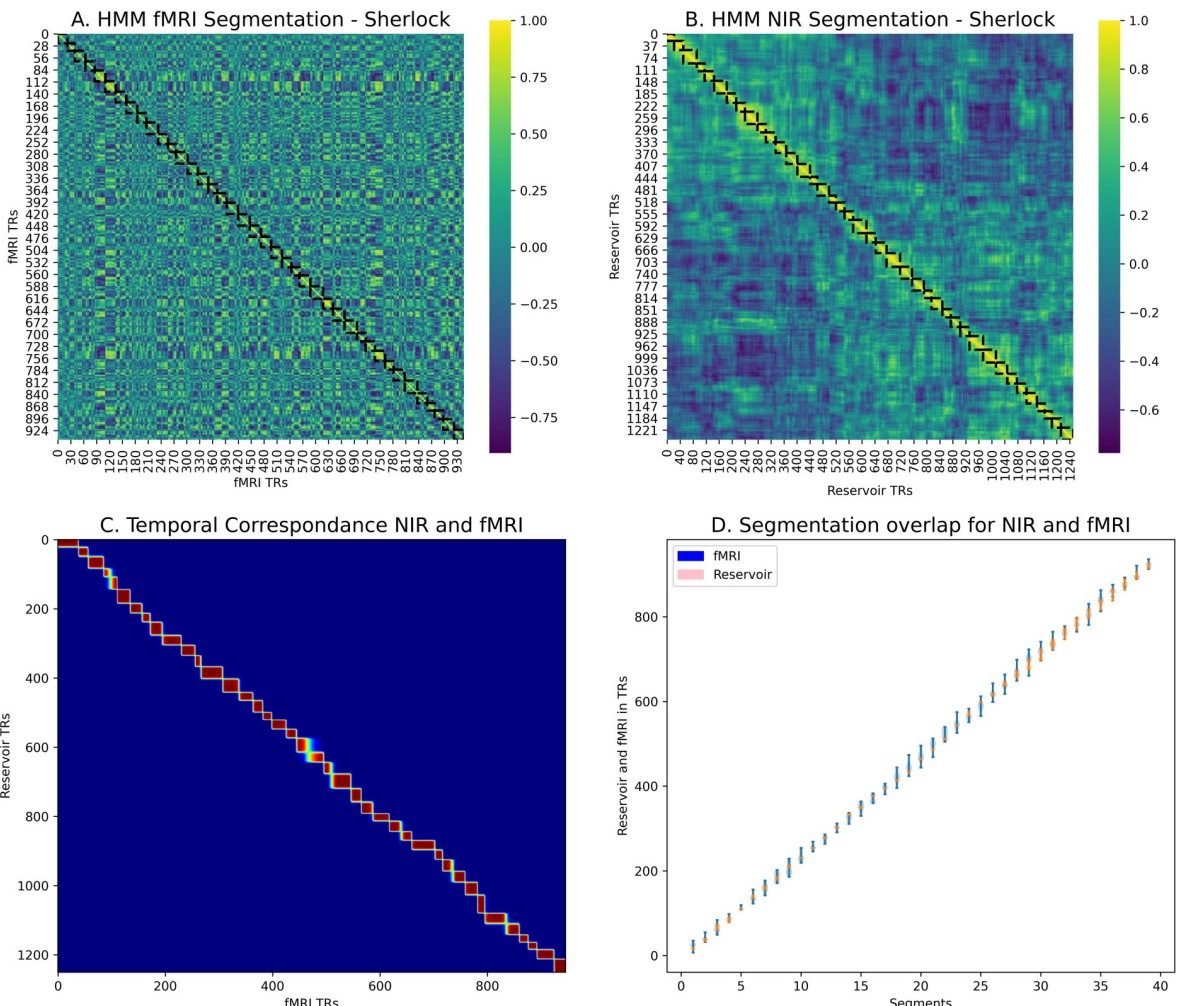

**Fig 4. Human and Model HMM segmentation.** A: Segmentation of human fMRI. B: Segmentation of Narrative Integration Reservoir model internal states. C: Temporal correspondence of fMRI and model segmentation. D: Correspondence of segmentation boundaries for fMRI from 16 subjects (blue) and reservoir trajectories from 16 model instances (pink).

dimensional embedding was retrieved and input into the reservoir. The HMM was then run on each of these reservoir activation trajectories. In Fig 4B we see the segmentation into 40 events in the average over all reservoirs.

In order to visualize the temporal correspondence between the reservoir and the fMRI, we can calculate the probability that reservoir state and an fMRI TR are in the same event (regardless of which event it is). This is

$$\sum k(T_R = k) \cdot p(T_B = k) \tag{1}$$

which we can compute by a simple matrix multiplication on the two segmentation matrices. (See BrainIAK Tutorial: https://brainiak.org/tutorials/12-hmm/). The result is illustrated in Fig 4C which illustrates the temporal correspondence between the segmentation of human fMRI and Narrative Integration Reservoir state trajectories. In Fig 4D we illustrate violin plots for the mean segment boundaries for the fMRI and NIR model. We can observe that the

distribution of segment boundaries for the NIR model spatially overlaps those for the fMRI data, indicating a good match.

To determine whether this match is statistically significant, we use the randomization procedure from Baldassano et al. [8] described above. We collected the segmentations for the 16 reservoir instances, and generated mean boundaries. We generate 1000 permuted versions of these NIR model boundaries in which the distribution of event lengths (the distances between the boundaries) is held constant but the order of the event lengths is shuffled. We use this null model for comparison with how often the HMM boundaries for the fMRI segmentation and those for the NIR model segmentation boundaries will be within 3 TRs of each other by chance. The true matches are significantly different from the null matches (p = 0.001), indicating that it is unlikely that the observed match is the result of random processes, allowing us to reject the null hypothesis. We performed the same test using the embeddings alone and the linear integrator. The linear integrator produced a segmentation that matched that of the fMRI (p = 0.0391). The embeddings alone produced a segmentation that did not differ from the null model (p = 0.1046).

The important point is that the Narrative Integration Reservoir demonstrates structured representations of narrative in terms of coherent trajectories of neural activity that are discontinuous at event boundaries, as revealed by the HMM segmentation. This allows us to proceed with investigation of temporal aspects of this processing.

## Different timing of constructing and forgetting temporal context

In order to investigate the time course of context processing, we exposed the reservoir to an experimental manipulation based on that used by Chien and Honey [9]. We recall that they considered two types of contextual processing: constructing and forgetting. In the constructing context, separate groups of subjects heard different narratives and then at a given point began to hear the same narrative. At this point, they began to construct a shared context. Conversely, in the forgetting context, the two separate groups of subjects initially heard the same narrative, and then at a given point began to hear two different narratives. At this point, they began to forget their common context.

We thus exposed paired instances of the Narrative Integration Reservoir (i.e. two identical instances of the same reservoir) to two respective conditions. The first instance was exposed to an intact version of the Not the Fall transcript in four components ABCD. The second instance was exposed to a scrambled version of the transcript ACBD.

For the two model instances, the initial component A is the same for both. The second and third components BC and CB, respectively, are different, and the final component D is the same. We can thus examine the transition Same to Different for forgetting, and the transition Different to Same for constructing. We expose two identical copies of the same reservoir separately to the intact and scrambled conditions, and then directly compare the two resulting reservoir state trajectories by subtracting the scrambled from the intact trajectory. This is illustrated in Fig 5. There we see that in the common initial Same section, there is no difference between the two trajectories. At the Same to Different transition we see an abrupt difference. This corresponds to forgetting the common context. Then in the transition from Different to Same, we see a more gradual convergence of the difference signal to zero in the construction context. Interestingly the same effects of abrupt forgetting and more progressive construction are observed for the linear integrator. This indicates that this asymmetry in constructing and forgetting is a property of leaky integrator systems.

This asymmetry with a rapid rise and slow decay in the values in Fig 5 may appear paradoxical, as integrators that are slow to forget prior context, should also be slow to absorb new

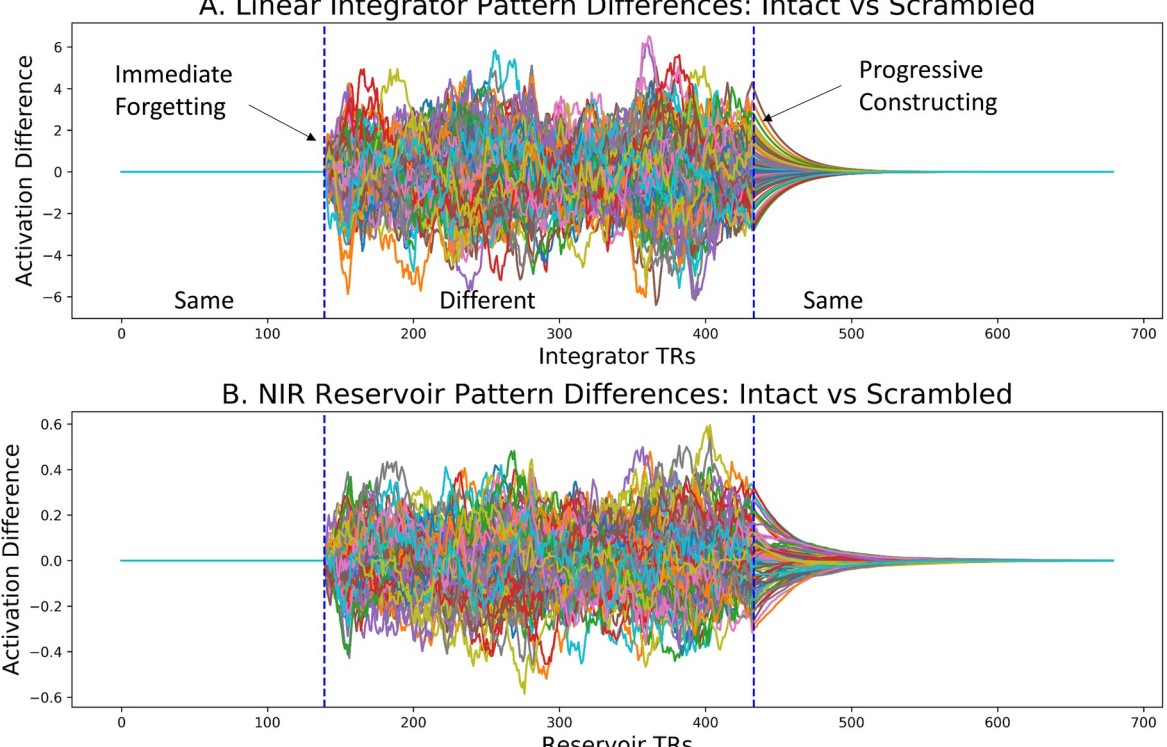

**Fig 5. Activation differences for model pairs exposed to intact and scrambled narrative.** A. Linear integrator and B. Reservoir activation difference at the Same-Different and Different-Same transitions. In the initial Same period, both perceived the same initial 140 words of the "Not The Fall" story. In the following Different period, model 1 continued with the story, and model 2 received a scrambled version. In the final Same period, both were again exposed to the same final portion of the story. Forgetting, at the Same-Different transition, takes place relatively abruptly. Constructing, at the Different-Same transition appears to take place much more progressively.

input. Thus it is important to recall that the dependent variable illustrated in Fig 5 is not the activation of a single integrator (or reservoir). Rather, it is the difference between two integrators (or reservoirs) exposed to the intact vs. shifted narratives, respectively. At the transition from same to different, the input to the two integrators becomes different, and from that point their pasts begin to differ as well. Thus, the integrator values in the same-different transition are dominated by the diverging inputs. This produces the rapid change in the difference value. In contrast, at the transition from different to same, the inputs become the same. In the two integrators only the pasts are different, and they converge to the same signal (driven by the common input) as a function of the leak rate. Thus, integrator values in the different-same transition are dominated by the divergent pasts as a function of the leak rate. These respective impacts of input and past produce the asymmetry in Fig 5. A more detailed analysis and demonstration is provided in the Materials and Methods section.

To analyze construction and forgetting across the two groups, Chien and Honey [9] measured the inter-subject pattern correlation (ISPC) by correlating the spatial pattern of activation at each time point across the two groups. We performed the same correlation analysis across the two reservoirs. In Fig 6 we display the forgetting and constructing context signals, along with the pairwise correlation diagrams, or inter reservoir pattern correlations IRPCs, formed by the correlation of states at time t of the intact and scrambled reservoir trajectories. Gradual alignment or constructing is illustrated in Fig 6A which shows the difference between reservoir trajectories for intact vs. scrambled inputs at the transition from Different to Same.

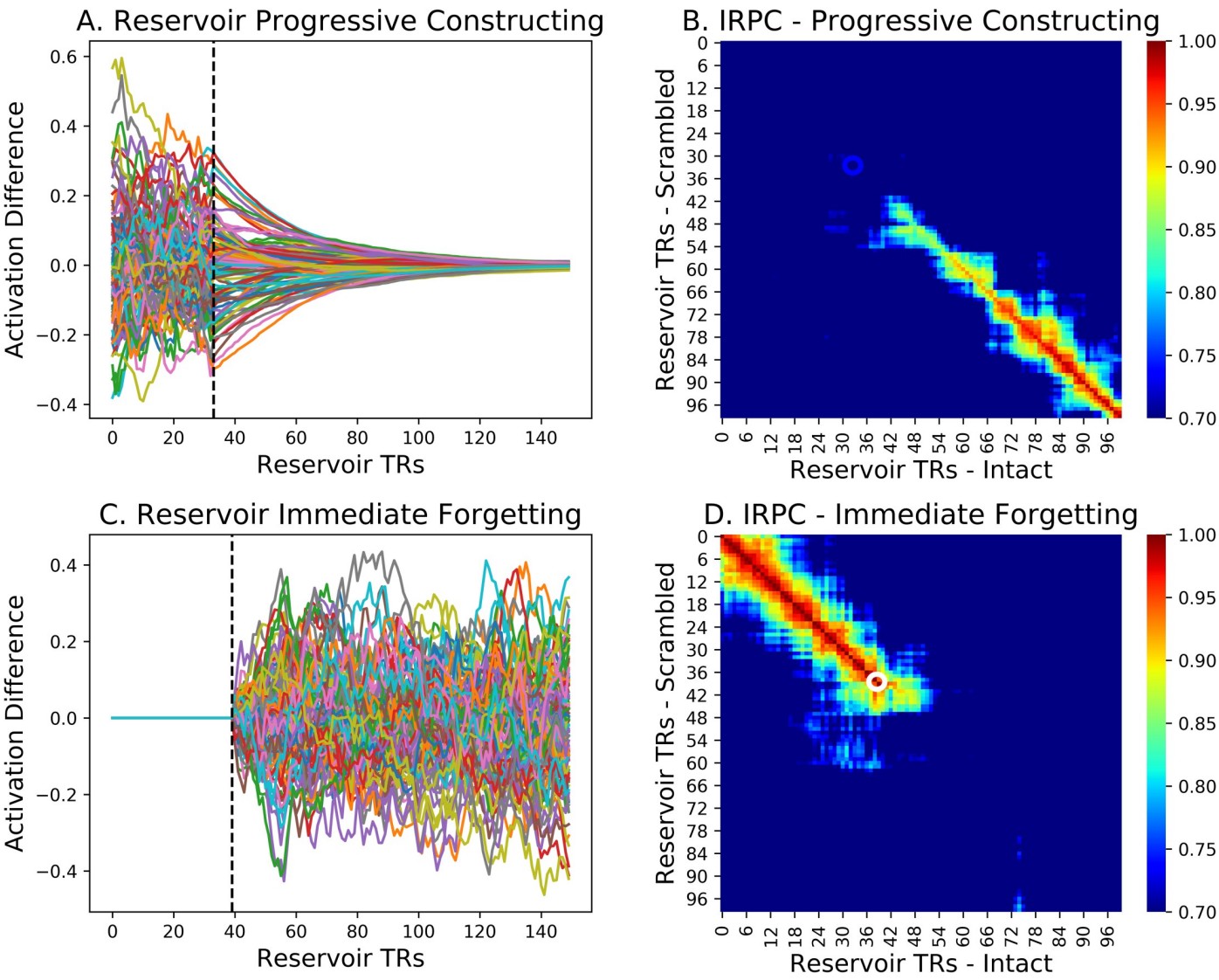

**Fig 6. Dynamics of Constructing and Forgetting.** A. Zoom on reservoir activity difference during constructing (transition from Different to Same). Dotted line marks the transition. Note the gradual decay. B. Time-point time-point correlations between the two (intact and scrambled input) reservoirs. Blue circle marks the beginning of the Same input. Note the slow and progressive buildup of coherence revealed along the diagonal. C. Zoom on reservoir activity difference during forgetting (transition from Same to Different). Dotted line marks the transition. Note the abrupt increase. D. Same as B, but white circle marks the Same to Different transition. Not the abrupt loss of coherence along the diagonal.

In Fig 6B the IRPC between the intact and scrambled reservoir state trajectories is illustrated, with a blue circle marking the point illustrated by the dotted line in 6A, where the scrambled and intact narratives transition to the common Same ending. Interestingly, we see that there is a gradual smooth reduction of the differences in 6A, and progressive re-construction of correlation along the diagonal in 6B. In contract, in 6C and D we focus on the transition from Same to Different. There we see the difference signal and the cross-correlation map for the divergence/forgetting context in the Same to Different transition. Presented in the same timescale as A and B, we see in C and D a much more abrupt signal change, both in the difference signal (6C) and in the IRPC map (6D). There, where the white circle marks the shift from common to different narrative input, we see an abrupt end to the correlation indicated by the initially

high value along the diagonal which abruptly disappears. Interestingly this is consistent with observations of Chien and Honey [9], who identified the existence of more extended time constant for constructing vs. forgetting a context in certain higher order cortical regions. Similar effects, were observed for the linear integrator. The take home message of Figs 5 and 6 is that in the forgetting and construction are not systematically related, and that forgetting produces a rapid divergence between the reservoirs' activations, while construction of a common context takes place over more progressively. We now examine these effects in more detail.

## Distribution of time constants for context Alignment/Construction

In Chien and Honey [9] this extended time constant for constructing was observed particularly for cortical areas higher in the semantic processing hierarchy such as the temporal-parietal junction TPJ, vs. primary auditory cortex. Indeed, they demonstrated while construction or alignment times increased from peripheral regions toward higher-order regions, forgetting or separation times did not increase in this systematic manner.

Considering such properties in the Narrative Integration Reservoir, we recall that the reservoir is made up of leaky integrator neurons with a leak rate, and thus the reservoir should have some characteristic inherent temporal dynamics and time constants. More importantly, within the reservoir, combinations of recurrent connections can construct subnetworks that have different effective time constants. This predicts that we should be able to identify a distribution of time constants within the reservoir, similar to the observations of Bernacchia et al. [29]. To test this prediction, we again exposed paired reservoir instances to the intact and scrambled conditions, and analyzed the difference between the two reservoir state trajectories. For each neuron in the reservoir pair, we took the absolute value of its activation difference (intact–scrambled) at the onset point of the convergence/construction period (indicated by the dotted vertical line in Fig 6A) and counted the number of time steps until that value fell to ½ the initial value. This was used as the effective time constant, or alignment time, of the construction rate for each neuron. The sorted values of these time constants is presented in Fig 7 where it compares well with the same type of figure illustrating the distribution of cortical construction time constants from Chien and Honey [9] (see their S4B Fig). When we applied this procedure to the linear integrator, we observed that all units have the same time constants. The distribution of time constants is a property of the reservoir that is not observed in the linear integrator.

We can further visualize the behavior of this distribution of time constants by binning neurons into time constant groups, forming different virtual cortical areas. Fig 8 illustrates two thus created virtual areas of 100 neurons each (neurons 100–199 for the fast area with fastest

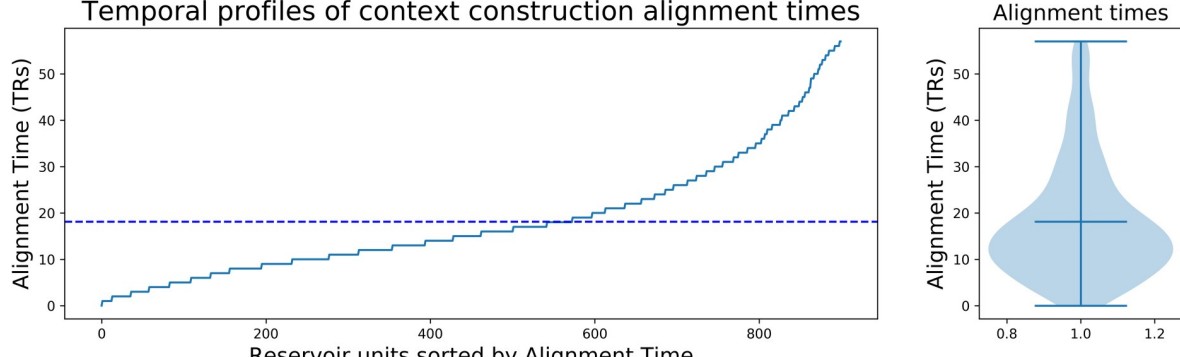

**Fig 7. Reservoir units sorted by alignment time constant (time steps for the activation difference to descend to ½ its initial value).** Broad range of alignment times. This distribution is remarkably similar to that observed by [9] (see their S4B Fig).

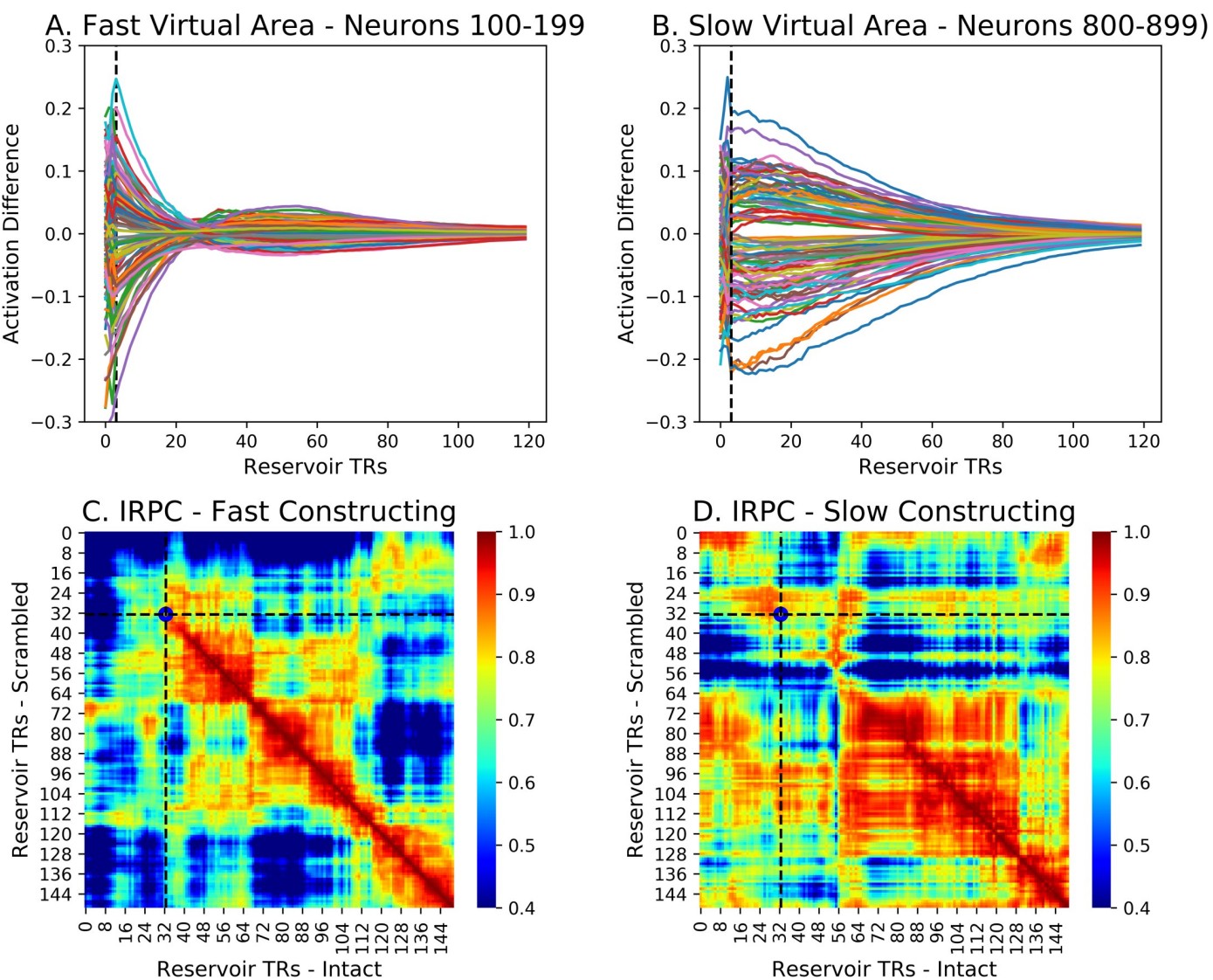

**Fig 8.** Effects of Alignment Time on Reservoir unit activity difference during alignment or constructing after the Different to Same transition, for fast (A) and slow (B) virtual area. Note the steep slope of the activity in the panel A that progressively flattens in panel B. Likewise note the "rebound" in A, where neurons quickly reduce their activity but then continue and cross the x-axis before coming to zero. In panel B, corresponding to slower neurons, activity actually increases at the transition before slowly coming back to zero. This indicates the complex and non-linear characteristics of the reservoir. C and D: Effects of alignment time on context construction. Same format as Fig 6B. Note how the rapid onset of coherence along the diagonal for the fast area in C, and the slower onset of coherence in the slower area in panel D.

time constants, and neurons 800–899 with slow time constants for the slow area). Note the slopes of the neuronal response which become more shallow in B vs. A. Below the traces of neural responses, the corresponding IRPC cross-correlation between the intact and scrambled reservoir state trajectories for each of these sub-groups of neurons is illustrated. We observe that for the faster neurons, the alignment is faster as revealed by the coherent structure along the diagonal.

Fig 9 illustrates these forgetting and constructing responses (panels A and B), and the inter reservoir pattern correlation functions for five increasing slow areas (made of 5 groups of 200 neurons) in the forgetting and constructing contexts (panels C and D). These linear plots correspond to the values along the diagonal of the IRPC—the inter reservoir pattern correlation

# Inter-Reservoir Pattern Correlations for Temporal Sub-Areas

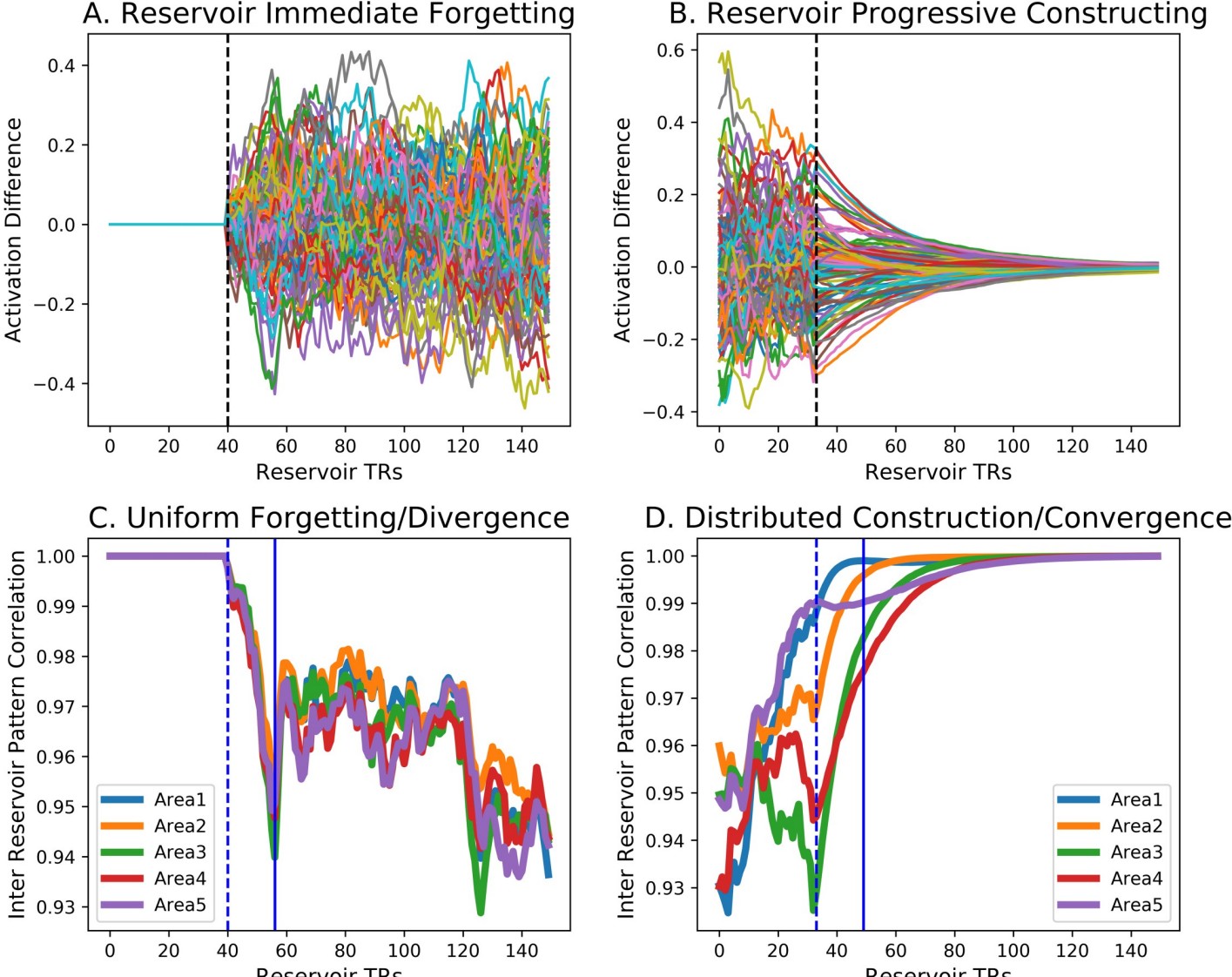

**Fig 9. Temporal profiles of Forgetting and Constructing.** A and B: Difference between reservoir activation for two NIR models receiving the intact and scrambled narrative. A. Forgetting—Dotted line indicates transition from Same to Different, with abrupt transition from 0 to large difference values. B. Transition from Different to Same, with transition from large differences, progressively to 0. C and D Inter-Reservoir Pattern Correlations for the two (intact and shifted input) NIR models. These linear plots correspond to the values along the diagonal of the IRPC. C. Forgetting. Dotted line indicates transition from Same to Different. All 5 temporal areas (Area1-5) display an overlapping descent to the minimum coherence in parallel. It reaches a minimum about 15 time steps later and then fluctuates around the same level. D. Constructing. In contrast, the 5 regions display a diversity of time-courses in their reconstruction of coherence. Dotted lines indicate the same time points in panels A-C and B-D respectively.

for the pairs exposed to the intact and scrambled narratives. There we observe that while the time constant for forgetting (transition from Same to Different) is fixed across these five cortical areas, there is a clear distinction in the time course of constructing (transition from Different to Same). Thus we observe a distribution of construction times, with no relation to the forgetting time. We verified the lack of correlation between the time constants for constructing vs. forgetting (Spearman correlation = -0.004, p = 0.89). This corresponds to the observations

of Chien and Honey [9], that alignment or construction time increases in higher cortical areas, whereas there is no systematic relation between cortical hierarchy and separation or forgetting [9]. Now we consider the functional consequences of this diversity of time constants in construction.

## Segmentation granularity corresponds to construction time constant

Baldassano et al. [8] showed that segmentation granularity varies across cortex, with shorter events in sensory areas and increasingly longer events in higher cortical areas. Chien and Honey [9] further revealed that the time constant for context constructing similarly increases along this hierarchy. Our time sorting method allows us to generate populations of neurons that can be grouped into virtual areas that are fast, that is they have a small effective time constant (i.e. high leak rate), and areas that are slow, with longer effective times constants (low leak rate). We can now test the hypothesis that there is a relation between these time constants for construction, and the increasing-scale of event segmentation effects in higher areas reported by Baldassano et al. [8]. In other words, longer time constants for construction will correspond to a preference for fewer, longer events in segmentation by the HMM, whereas shorter time constants will have a preference for more numerous and shorter events in the segmentation by the HMM.

We predict that the HMM will have a better fit when k is smaller for a slow area, and when k is higher for a fast area. Formally for values of k = i, and k = j (denoted $k_i$ and $k_j$, we test the following inequality, when i < j, and thus $k_i < k_j$: the sum of log-likelihoods for small k and slow NIR, and large k and fast NIR will be greater than the sum for large k and slow NIR and small k and fast NIR. We refer to this difference as the segmentation effect.

$$for\ i < j,\ ll(k_i(NIR_{slow})) + ll(k_j(NIR_{fast})) > ll(k_j(NIR_{slow})) + ll(k_i(NIR_{fast})) \qquad (2)$$

To test this prediction, we ran the forgetting-construction experiment as described above, and then generated fast and slow virtual cortical areas of 100 neurons each, corresponding to those illustrated in Fig 8, and segmented the activation from these fast and slow areas using the HMM with large and small values of k. We then evaluated the prediction as the inequality in Eq (2). We first consider the results with the HMM with $k_i = 8$ for the slow area, and $k_j = 22$ for the fast area for an example reservoir, illustrated in Fig 10A and 10B. These values for k were identified based on specification by Baldassano et al. [8] of optimal values of k for fast (early visual) and slow (default mode) cortical areas (see Materials and Methods). The results using these values of k for segmenting the NIR neural activity are illustrated in Fig 10A with the time-point time-point correlation map for a fast area (neurons 100–199) with small convergence time constants, and the event boundaries found by the HMM with k = 22. Panel B illustrates the same for a slow areas (neurons 800–899) with large convergence time constants, and the event boundaries found by the HMM with k = 8. We can observe the finer grained correlation structure along the diagonal in panel A, and more coarse grained, larger event structure in panel B. For 50 NIR instances, this segmentation effect was significant, p<0.001, consistent with our prediction. That is, the sum of log-likelihoods for k = 8 for the slow NIR, and large k = 22 for the fast NIR was significantly greater than the sum for k = 22 with the slow NIR and k = 8 for the fast NIR.

To examine the more general application of the inequality in Eq (2) we ran the HMM on these two areas, iterating over k from 2 to 40 for both areas. The results of this grid search are presented in Fig 10C and 10D, which illustrate the value of the inequality, and the p values of the difference, respectively. There see we that for all small values of $8 < k_i < 16$ applied to the slow area and large values $18 < k_i < 40$ applied to the fast area, the inequality in Eq (2) holds,

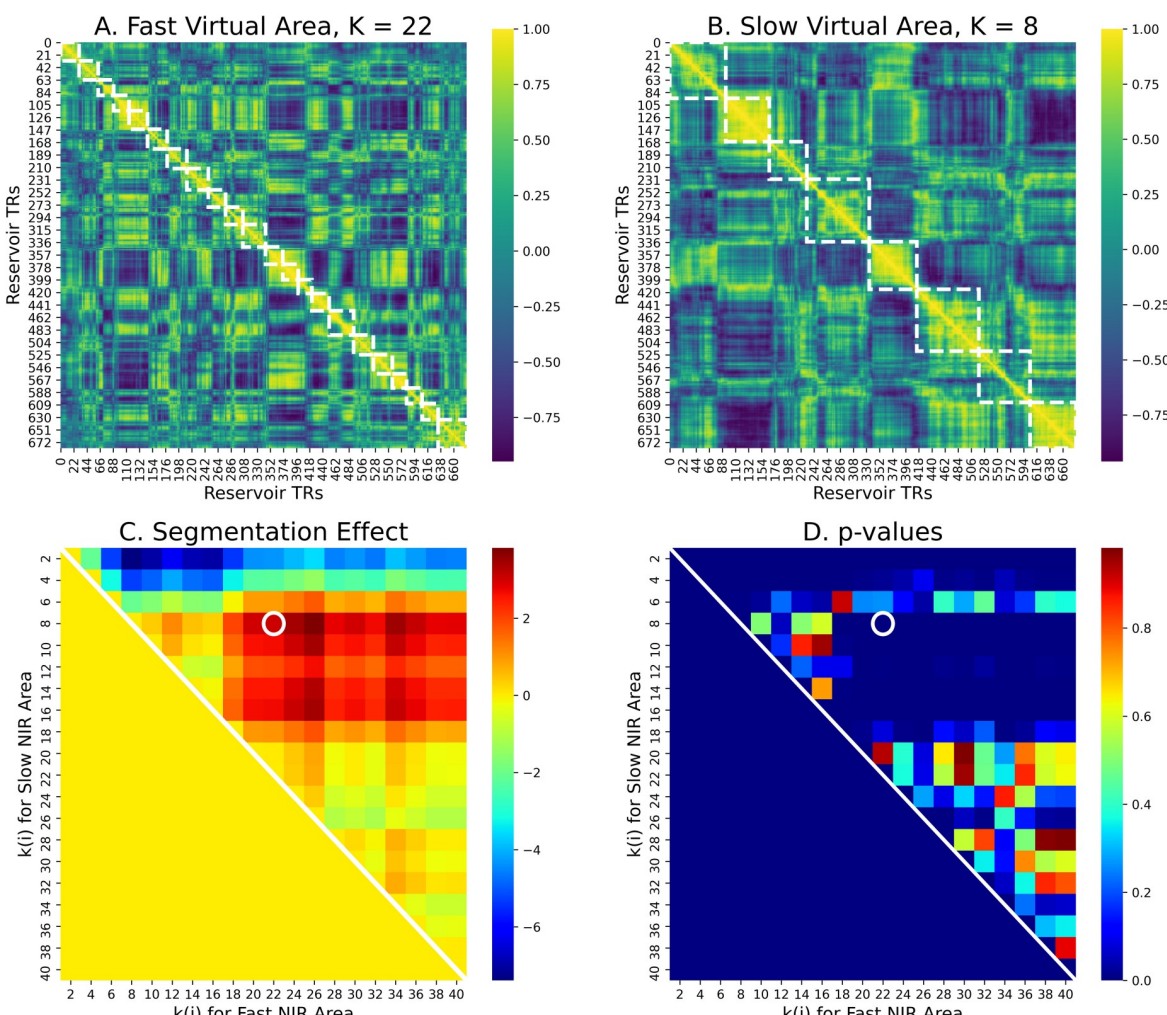

**Fig 10. HMM segmentation in two populations of reservoir neurons with faster vs. slower alignment times.** A–Fast reservoir subpopulation (neurons 100–199) with shorter alignment times. HMM segmentation into k = 22 event segments. B–Slow reservoir subpopulation (neurons 800–899) with greater alignment times. HMM segmentation into k = 8 regions. Note the larger sections of continuous coherence in B vs. A. Panel C—Segmentation effect advantage for 50 reservoir pairs for using k(i) < k(j) on the slow vs fast areas, respectively. D. p values. For k(i) = 8–18 and k(j) = 18–40 this advantage is significant. For a large range of k values, when k(i) < k (j), there is a significant advantage of using the HMM with k(i) vs. k(j) on the slow vs. fast areas respectively.

with median p = 0.0001, mean p = 0.0015. This corresponds to the red patch above the diagonal in Fig 10C for $8 < k_i < 16$ and $18 < k_i < 40$, indicating the positive segmentation effects, and the corresponding blue patch in Fig 10D, corresponding to the significant p values. This confirms the prediction that areas with faster construction times prefer shorter events, and those with slower construction times constants prefer longer events.

## Discussion

Narrative is a uniquely human form of behavior that provides a coherent structure for linking events over time [6,48]. A given instance of a narrative itself is a temporal sequence that must be processed by appropriate mechanisms in the nervous system [10]. Previous research has identified constraints on this processing related to immediacy [38]. Information that is provided early in the narrative must remain immediately accessible at any time. This is technically

an interesting problem, as in many architectures, the time to access previous memories may depend on the size of the memory, or the position of the data in question in the input sequence. We recently reasoned that performing a temporal-to-spatial transformation can solve this problem, provided that there is a parallel readout capability that provides immediate access to the distributed spatial representation [36].

The current research further tests this model in the context of temporal processing constraints identified in more extended narrative processing. Chien and Honey [9] compared brain activity across subjects that hear the same and then different narratives (forgetting), and different then same narratives (constructing). They characterized a form of asymmetry in the temporal processing of the construction and forgetting of narrative context. Construction and forgetting can take place at different rates, and particularly in higher areas, construction can take place more progressively than forgetting. This is our first constraint. More globally, their empirical finding was that while construction or alignment times increased from peripheral regions toward higher-order regions, forgetting or separation times did not increase in this systematic manner. This gradient of construction times is our second constraint.

In order to account for the hierarchy of time constants across areas Chien and Honey [9] developed the HAT model with explicit hierarchically organized modules, where each successive module in the hierarchy is explicitly provided a longer time constant. They introduced a gating mechanism in their hierarchical recurrent model in order to explicitly address the more abrupt response to change in context in the forgetting context. This model is a significant achievement as it provides an explicit incarnation of the processing constraints in the form of a hierarchy of modules, the pre-specified time constants and the explicit gating mechanism. As stated, the problem remains, "what are the essential computational elements required to account for these data?" (p. 681–682). Our third constraint comes from Baldassano et al. [8] who demonstrate that higher elements in the cortical hierarchy prefer segmentation with fewer and larger segments, while lower sensory areas prefer more and shorter segments. Part of the motivation for the current research is to attempt to respond to the question "what are the essential computational elements required to account for these data?" with respect to these constraints, and to establish a relation between the context construction hierarchy of Chien and Honey [9] and the event segmentation hierarchy of Baldassano et al. [8].

Indeed, when we simulated the experimental conditions where pairs of subjects heard the same and then different narratives (forgetting) and respectively different and then same narratives (constructing), the between reservoir correlations displayed an abrupt response in the forgetting context, and a more progressive responses in the constructing context. Interestingly, this behavior is an inherent property of the reservoir, and the linear integrator, in the context of a task that evaluates the difference between two reservoirs (or integrators). The transition from different to same requires the dissipation of the past differences in face of common inputs, and will vary as a function of the leak rate. In contrast, in the transition from same to different, the rapid onset response in the forgetting context is due to immediate responses to the diverging inputs in the reservoir which are independent of the leak rate (see Fig 14 and Materials and Methods). This behavior of the reservoir and the linear integrator addresses the first constraint from Chien and Honey [9]–the construction, forgetting asymmetry, or lack of covariance between constructing and forgetting. However, whereas each element of the linear integrator has the same decay rate, the hierarchy of decay time constants is a natural property of the reservoir, as observed by Bernacchia et al. [29] (and see Materials and Methods below). Thus, only the reservoir, and not the linear integrator, can address the second constraint, the hierarchy of time constants across cortical areas for forgetting.

By partitioning the reservoir neurons into subgroups based on their constructing time constants, we observed that areas with longer time constants preferred segmentation with fewer,

longer events, while on the opposite, areas with shorter time constants preferred segmentation with more numerous and shorter events, thus addressing the third constraint, from Baldassano et al. [8]. This was demonstrated using evaluation of the preferred number of events based on the log-likelihood of model fit. New methods for event segmentation [49] can overcome limitations of this approach including a tendency to overestimate the number of states when the number of states approaches the number of time-points. As we are far from this regime (i.e. the number of states in much smaller than the number of time points) the HMM-based method is not subject to this problem, and suitable for our purposes [49].

While there are likely multiple mechanism that contribute to the diversity of inherent time constants across cortical areas [29,50,51], the internal dynamics of recurrent loops within the reservoir account for a rich diversity of effective time constants as we observe here. We propose that this segregation is a natural product of neuroanatomy, and that narrative structure has found its way into this structure. While individual neurons have the same time constants, we observe a recurrent time constant effect, a functional variety of time constants due to the recurrent dynamics.

Here we observed how this effect generates a diversity of time constants within a single reservoir. We partitioned the units into groups based on time constants, and revealed a relation between these time constants for context construction, and event segmentation across cortical areas as observed by [8]. This effect should also extend across multiple areas in a larger scale model, with a hierarchy of time constants as one moves farther from the input, consistent with Chaudhuri et al. [51].

We propose that in the primate brain, as higher cortical areas become progressively removed from sensory inputs, and driven by lower cortical areas, they will have higher effective time constants, due to the accumulation of the recurrent time constant effect. The result would be a local diversity of time constants within a given range within areas, consistent with [29], and a broader range of time constants across areas consistent with [50] and [51]. The potential results of such effects have been observed in the hierarchy of temporal receptive windows in cortex [52,53] and the corresponding temporal processing hierarchy [8,9,52]. Lerner et al. [53] thus characterized a hierarchy of temporal response windows by examining inter-subject correlations in brain responses to audio that was intact, or scrambled at word, sentence or paragraph levels. It will be interesting to test the prediction that the virtual areas in the reservoir with their different alignment times will also display corresponding temporal response windows as revealed by scrambling at different levels of granularity as observed by [53]. Here we propose that the different levels of temporal structure in narrative may have evolved to be accommodated by the representational capabilities in the cortical hierarchy of temporal processing.

It is worth noting, in the context of the relation between narrative structure and recurrent network functional neuroanatomy, that the observed reservoir behavior comes directly from the input-driven dynamics of the reservoir. There is no learning in the reservoir, and we do not look at the readout, but at the recurrent elements of the reservoir itself. While there is no learning in the reservoir, the NIR model benefits from massive learning that is present in the Wikipedia2Vec corpus [39]. Indeed, the embeddings themselves already contain sufficient information to provide segmentation. Words within the same narrative context will tend (by definition of the embedding model) to share similarity with words in the same narrative context. The benefits of this learning are clearly visible in the visible structure of the embeddings in Figs 2 and 9, and in the segmentation results that can be obtained directly from the embeddings and from a simple linear integrator model that we examined.

Indeed, comparing the reservoir to the embeddings alone and the linear integrator reveals the crucial features of the reservoir, which derive from its highly non-linear integrator

properties. That is, the distribution of effective time constants that are found within the reservoir, and that emerge because of the recurrent connectivity. The recurrent connectivity provides a diversity of effective time constants. Grouping neurons by these time constants allows us to evaluate the hypothesis that effective functional time constants in cortical areas as observed by Chien and Honey [9] will correspond to granularity of event representations in these areas, as revealed by HMM segmentation of Baldassano et al. [8]. This corresponds to the added-value of the reservoir with respect to a linear integrator as a process model of cortical dynamics. The recurrent dynamics provide two key elements of cortical function that are not present in the linear integrator. The first, which we have previously examined, is the projection of the inputs into a high dimensional space, which provides the universal computing characteristic characterized by [23], and revealed by mixed selectivity in the neuronal coding [18,22]. Here we see how this high dimensional coding models cortical integration during narrative processing. The second property, which we investigate for the first time here, is the diversity of functional time constants that is a correlate of the high dimensional projection. Indeed, in this high dimensional representation, time is one of the dimensions [31]. The degree of resemblance that we found between cortical and reservoir dynamics in narrative processing provides further support for the idea that aspects of cortical function can be considered in the context of reservoir computing [18,19,21,22,54].

A limitation of the current modeling effort is that it does not explicitly address meaning and the content of the representations in the NIR model and the fMRI signal. Indeed, in our previous work [36], we predicted N400 responses using the trained readout of the reservoir to generate the cumulative average of the input word sequence, thus forming a discourse vector. This discourse average vector could then be compared with the word embedding for a target word in order to predict the N400 as 1-similarity. Interestingly, these discourse vectors encode knowledge that is assumed to be required for making inferences about events [42]. Related models have used recurrent networks to integrate meaning over multiple word utterances in order to predict the N400 [55,56], further supporting the role of recurrent connections in accumulating information over multiple words. Reservoir content can also be used to decode more structured meaning about events in terms of semantic roles including agent, action, object, recipient etc. [35,57]. This information is coded in the input based on clues provided by grammatical morphology and word order [58], in the form of grammatical constructions [59,60]. In the reservoir model, these inputs are combined in the high dimensional representation, allowing a trained readout to extract semantic structure. The same principal should apply to narrative structure. That is, the elements contributing to the narrative meaning in the input are combined in the high dimensional representation, and a trained readout should be able to extract the desired meaning representations. Related research has started to address how structured meaning in the narrative is extracted to build up a structured situation model [61,62], so that the system can learn to answer questions about perceived events and narrative. Baldassano et al. [8] note that the event segmentation they observe in high areas including angular gyrus and posterior medial cortex exhibit properties associated with situation model representations including "long event timescales, event boundaries closely related to human annotations, generalization across modalities, hippocampal response at event boundaries, reactivation during free recall, and anticipatory coding for familiar narratives" (p. 717). Baldassano et al. [63] further investigated these representations and determined that the event segment structure discovered by the HMM can be used to classify fMRI activation trajectories based on the underlying story schema.

In this context, we are witnessing an interesting conjuncture in the science and technology of language. Language models in machine learning are beginning to display remarkable performance capacities with human-like performance in question answering [64], semantic similarity judgement, translation and other domains [65,66]. In certain aspects they are similar

enough to human performance that specific measures of human language comprehension from psycholinguistic experiments are now being used to characterize and evaluate these language models [67,68]. At the same time, these language models are beginning to display underlying representations and mechanisms that provide insight into human brain processes in language processing [69–71]. Future research should combine human neurophysiology studies of narrative comprehension and parallel modeling of the underlying neurophysiological processes. In this context one would expect to identify the presence of high dimensional coding and mixed selectivity, characteristic of reservoir computing, in cortical processing of narrative.

## Materials and methods

### Narrative integration reservoir

The end-to-end functioning of the Narrative Integration Reservoir (based on, and extending the model of Uchida et al. [36]) is illustrated in Fig 1. The model consists of two components. The first generates word embedding vectors, and the second generates the spatiotemporal trajectory of neural activation. Given the input narrative, we first remove stop words (e.g. the, in, at, that, which, etc.) which provide little or no semantic information [72]. The remaining input words are transformed into word embedding vectors by the Wikipedia2Vec model, pretrained on the 3 billion word 2018 Wikipedia corpus [39]. These vectors are then input to the reservoir, a recurrent network with fixed recurrent connections. The fixed connections in the recurrent network allow the full dynamics of the system to be exploited, which in some other networks with modifiable recurrent connections is not the case due to temporal cut-off of the recurrence required for implementing learning on the recurrent connections [12,27,73]. This method of fixed connections in the reservoir was first employed in order to model primate prefrontal cortex neurons during behavioral sequence learning tasks [19], and was subsequently developed with spiking neurons in the liquid state machine [23], and in the context of non-linear dynamics signal processing with the echo state network [24,74], all corresponding to the class of reservoir computing [25].

In reservoir computing, the principle is to create a random dynamic recurrent neural network, and then stimulate the reservoir with input, and harvest the rich high dimensional states. Typically this harvesting consists in training the output weights from reservoir units to output units, and then running the system on new inputs and collecting the resulting outputs from the trained system. In the current research we focus our analysis directly on the rich high dimensional states in the reservoir itself. That is, we do not train the reservoir to perform any transformation on the inputs. Instead, we analyze the activity of the reservoir neurons themselves. The basic discrete-time, tanh-unit echo state network with $N$ reservoir units and $K$ inputs is characterized by the state update equation:

$$\mathrm{x}(t+1) = (1-\alpha)\mathrm{x}(t) + \alpha \cdot f(\mathbf{W}\mathrm{x}(t) + \mathrm{W_{in}}\mathrm{u}(t)) \tag{3}$$

where $\mathbf{x}(n)$ is the $N$-dimensional reservoir state, $f$ is the tanh function, $\mathbf{W}$ is the $N{\times}N$ reservoir weight matrix, $\mathbf{W}in$ is the $N{\times}K$ input weight matrix, $\mathbf{u}(n)$ is the $K$ dimensional input signal, $\alpha$ is the leaking rate. The matrix elements of W and $W_{in}$ are drawn from a random distribution.

The reservoir was instantiated using easyesn, a python library for recurrent neural networks using echo state networks (https://pypi.org/project/easyesn/) [46]. We used a reservoirs of 1000 neurons, with input and output dimensions of 100. The W and W_in matrices are initialized with uniform distribution of values from -0.5 to 0.5. The leak rate was 0.05. This was established based on our empirical observations and the high volatility of the input. We also tested with leak rates of 0.1, 0.15 and 0.2. The reservoir is relatively robust to changes in these values, as long as the reservoir dynamics are neither diverging nor collapsing.

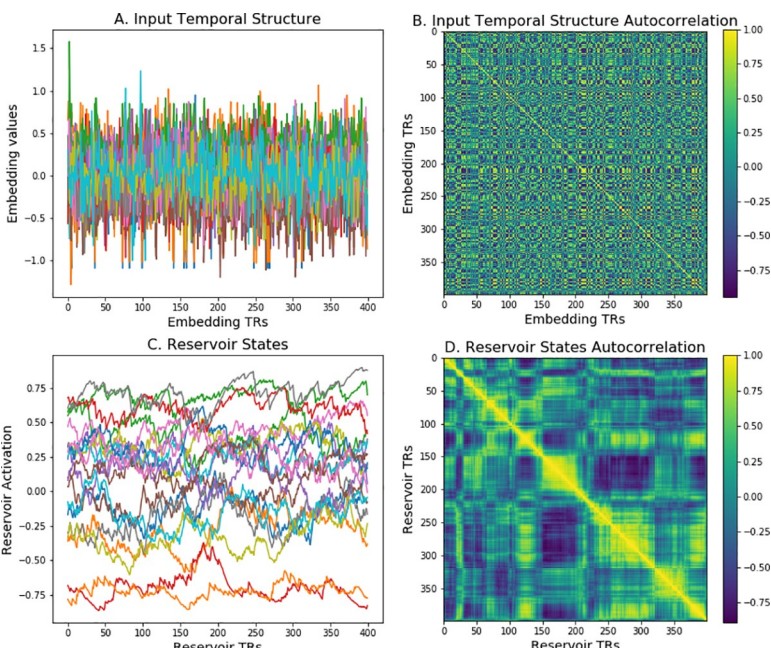

**Fig 11. Reservoir fundamentals.** A. Temporal structure of input sequence of word embeddings for 100 input dimensions. B. Time-point time-point pattern correlation for input sequence. C. Activity of a subset of reservoir units. Note the relative smoothing with respect to the input in panel A. D. Time-point time-point pattern correlation of reservoir activity during the processing of the narrative. Note the display of coherent structure along and around the diagonal, when compared with panel B. This indicates the integrative function of the reservoir.

To simulate narrative processing, words were presented in their sequential narrative order to the reservoir. Stop words (e.g. the, a, it) were removed, as they provide no semantic information [72]. Similar results were obtained in the presence of stop words. Words were coded as 100 dimensional vectors from the Wikipedia2vec language model. Fig 11 illustrates the forms of the input, and reservoir activation, and their respective cross-correlations.

In Fig 11, in panel A we see the high frequency of change in the input signal. This is the word by word succession of 100 dimensional word embeddings for the successive words in the narrative. In B we see the auto-correlation, and note that there is structure present. (Event segmentation for embeddings is presented in Fig 2). In C we see at the same timescale the activation of 50 reservoir units. Here we can observe that the frequency of change is much lower than in the original input. This is due to the integrative properties of the reservoir. In D we see the autocorrelation of the reservoir states over the time course of the narrative. Here we see along the diagonal more evidence of structure and the integration over time in local patches.

The reservoir has inherent temporal dynamics. We can visualize these dynamics by exposing the reservoir to a zero input, then a constant input, and then return to zero, and then observing the responses to these transitions. Such behavior is illustrated in Fig 12. A zero input is provided from 0 to 500 time steps, then a constant input from 500 to 900, and finally a zero input from 900 to 1500. Fig 12 displays the response of 10 sample neurons. This illustrates the inherent temporal dynamics of the reservoir. In order to more carefully characterize the temporal properties of the reservoir, we measured the time constants for neural responses to these transitions, and then plotted the ordered time constants. In Fig 13 we display the ordered time constants for neurons in response to the transition from zero input to a fixed non-zero signal, and then from signal to zero. These can be compared to the time constants for construction and forgetting.

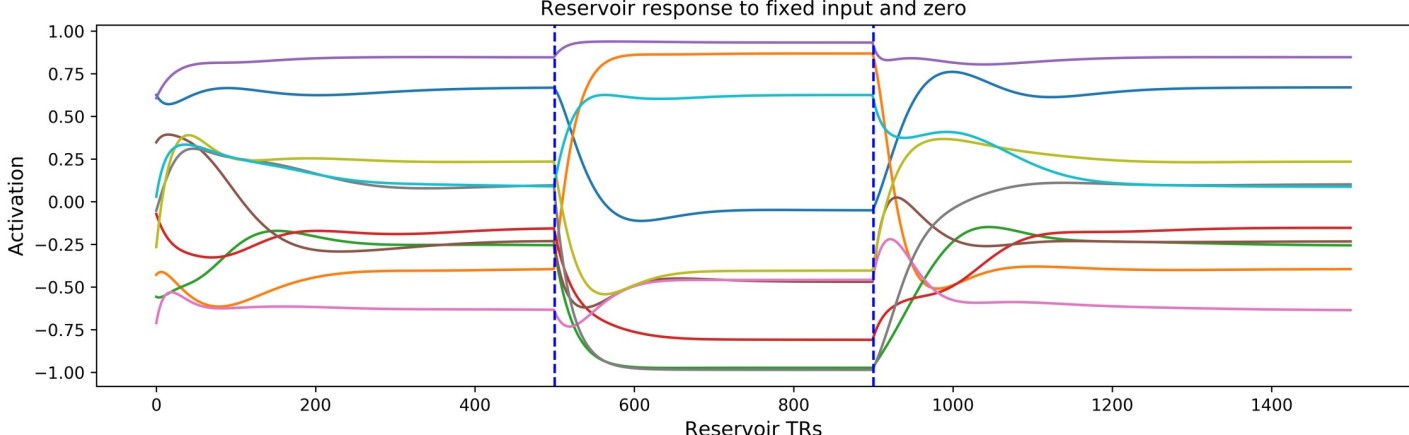

**Fig 12. Reservoir dynamics in response to a continuous input of zero, then a fixed non-zero input, and final return to zero input.** Note the diversity of temporal responses. This indicates the inherent property of distributed time constants generated by recurrent connections within the reservoir.

## Linear integrator

In order to demonstrate that the model's performance is different from feeding the embeddings into a linear integrator model, we use a linear integrator described in Eq (4):

$$LI_{(n)} = (1-\alpha)*LI_{(n-1)} + (1+\alpha)*embedding_{(n)} \qquad (4)$$

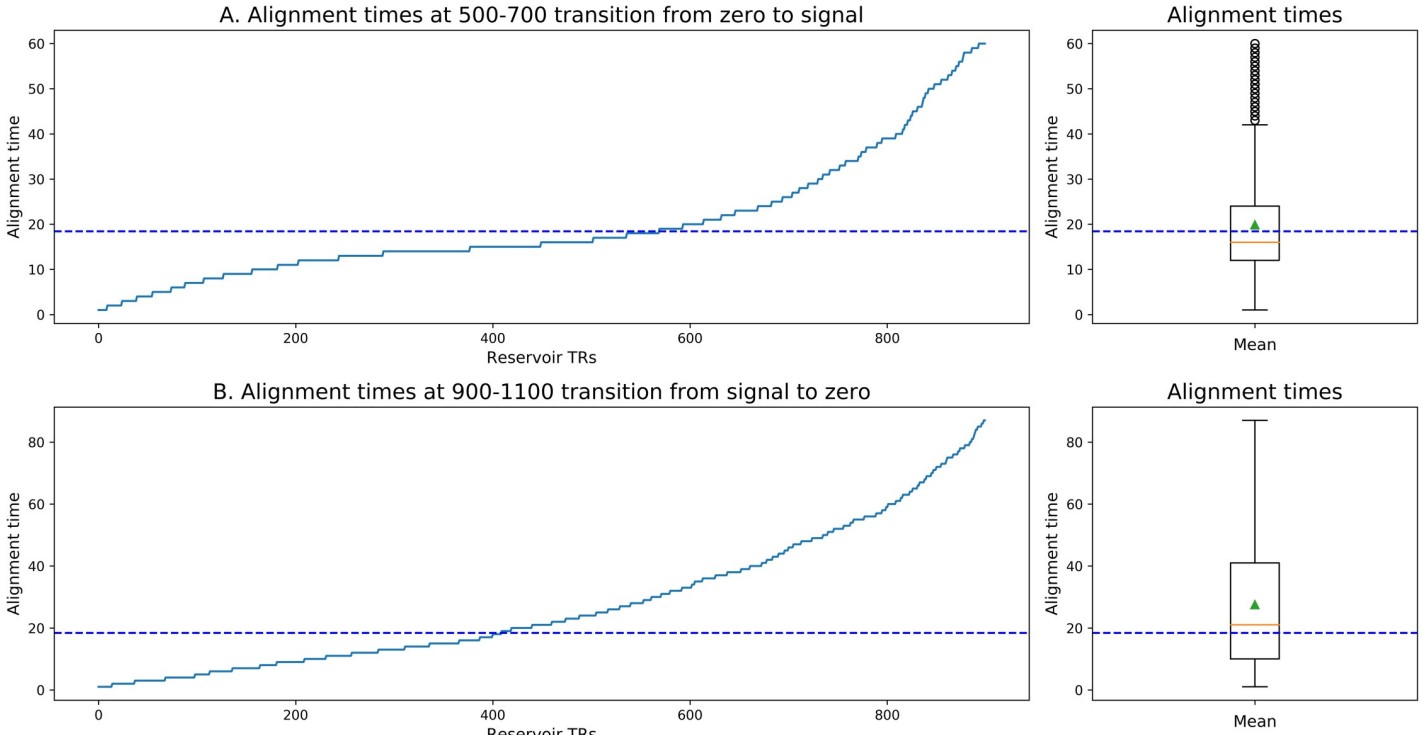

**Fig 13. Reservoir units sorted by time to stabilize after the transitions from zero to non-zero input, and from non-zero to zero input.** Comparable to Fig 7. Dotted line marks the mean from Fig 7 for comparison.

The linear integrator LI combines the previous integrated embeddings with the current embedding scaled by a leak rate α. As α increases, the influence of past inputs is reduced. In the different experiments we compare performance of the NIR model with the LI linear integrator.

## Note on the asymmetry for constructing and forgetting in the linear integrator

Here we analyze the behavior of the linear integrator described in Eq (4) in the context of the construction/forgetting asymmetry. We first confirmed that indeed the LI behaves as expected to a pulse input, with symmetric rise and fall behavior. Then, in order to determine that the asymmetry is not due to correlation in the input, we ran the intact vs. scrambled experiment for constructing and forgetting using data sampled from a random normal distribution. This is illustrated in Fig 14. Again we observe forgetting-construction asymmetry panels A and B.

We determined that the asymmetry is due to properties of the integrator in a certain parameter range, in the specific context of the current task. What is visualized in Fig 5, and 14, is not the response of a single instance of the integrator, but rather it is the difference in the activations of two integrators exposed to particular input sequences that have an ABCD and ACBD structure, respectively. Fig 14A and 14B display this difference between the 100 units of the two LIs for the Gaussian data. This is directly comparable to Fig 5A.

Panels C and D display this difference for a single example unit, with three different values of the leak rate ($\alpha = 0.2, 0.1, 0.05$). For forgetting, in panel C, we can observed that the difference value changes rapidly, and the leak rate does not have a large impact. For forgetting, at each time step the inputs to the two integrators are different. This contributes to the rapid divergence. In addition, the memory components integrate these differences. Thus, in the forgetting condition, the inputs and the memory components of the two integrators vary, leading to a rapid divergence, dominated by differences in the inputs. For construction, in panel D, we can observe that the difference value is initially high, and converges to zero at different rates, dependent on the leak rate. At each time step, the inputs are the same and only the memory components of the two integrators are different. We see the clear effect of the memory component, as in panel D, we observe that the leak rate plays an important role in the time-course of construction. This reveals that our observed asymmetry in forgetting vs. construction is due to the combined effects of (a) the task–taking the difference between two integrators in the same-different-same input conditions (as opposed to observing a single integrator), and (b) the leak rate of the linear integrators. At the same-different transition, the difference is driven by the inputs which diverge rapidly. At the different-same transition, the difference is driven by the memory component, which evolves more or less slowly, according to the leak rate.

In the linear integrator of Chien and Honey [9], it is likely that with relatively high leak rates (i.e. low vales of parameter $\rho i < 0.5$ in their Eq 1 describing the linear integrator), this asymmetry would not be present.

## Note on data normalization for correlation maps

When illustrating the time-time correlation maps, we do so after subtracting the mean value from each element (i.e., for each unit in the reservoir, or each dimension in the word-embedding, we treat the signal as a time-series, and subtract the mean of the time-series from every time-point). This can prevent saturation issues which may confuse the interpretation.

## BrainIAK HMM model

The HMM model is described in detail in Baldassano et al. [8] and is available as part of the BrainIAK python library, along with example jupyter notebook that corresponds to [8]

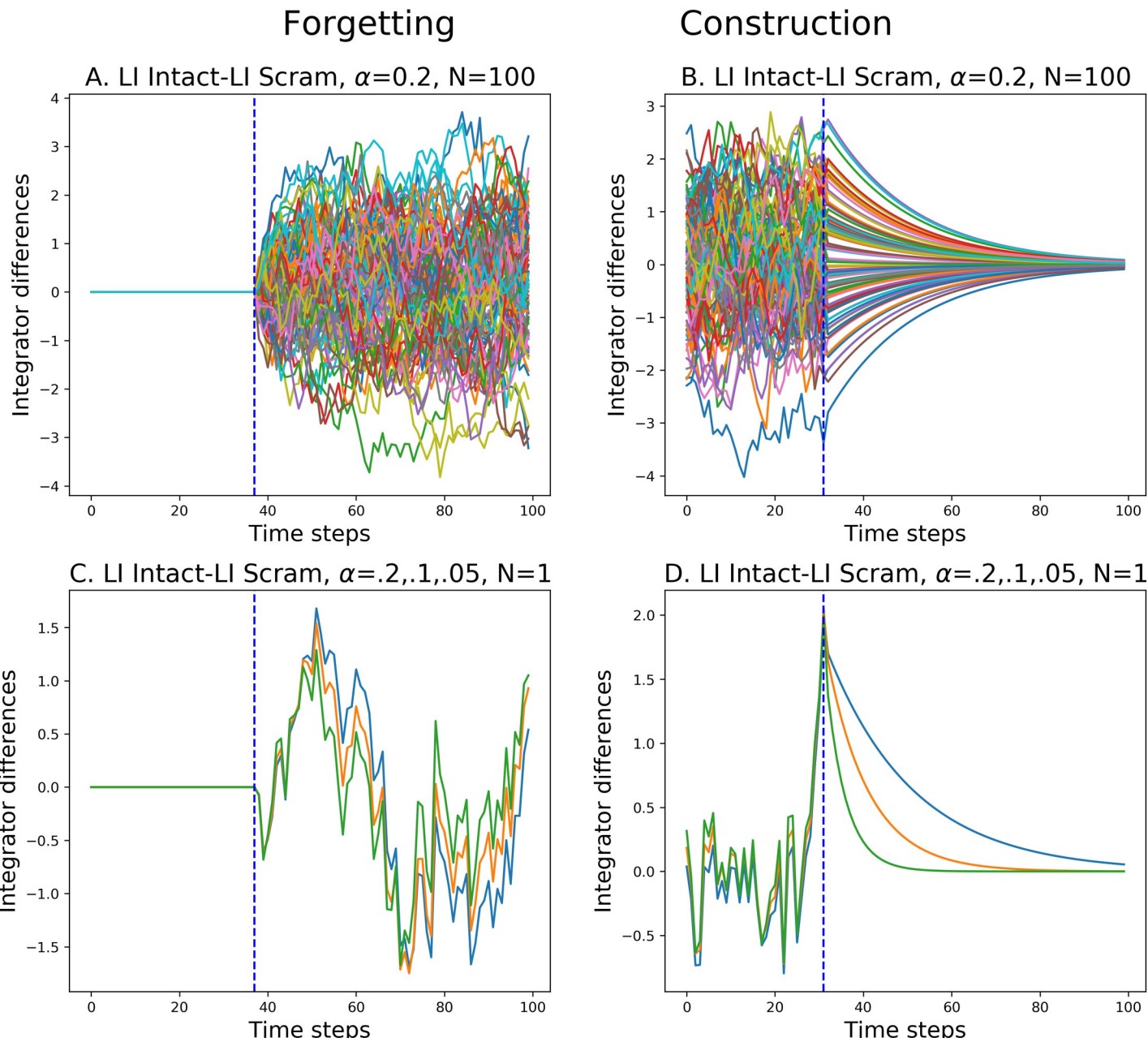

**Fig 14. Analysis of Linear Integrator in Forgetting and Construction.** Using random Gaussian inputs, intact (ABCD) and shifted (ACBD). A. Forgetting–transition from same to different inputs marked by dotted line. All 100 units of LI. Dominated by input term of LI. B. Construction–transition from different to same inputs marked by dotted line. All 100 units of LI. Dominated by memory term of LI. C-D. Example single unit of LI, tested with three time constants ($\alpha$ = 0.2, 0.1, 0.05 –blue, orange, green). C. Time constants do not have a remarkable effect on the difference signal for forgetting at the transition. D. Time constants have a noticeable effect on the difference signal for construction at and beyond the transition.

(https://github.com/brainiak/brainiak/tree/master/examples/eventseg). Given a set of (unlabeled) time courses of (simulated or real) neural activity, the goal of the event segmentation model is to temporally divide the data into "events" with stable activity patterns, punctuated by "event boundaries" at which activity patterns rapidly transition to a new stable pattern. The number and locations of these event boundaries can then be compared between human neural activity and simulated Narrative Integration Model activity, and to ground truth values.

The model is run on a given fMRI or reservoir trajectory, and requires specification of the expected number of segments, k. The segmentation returns a probability value. The trained model can then be run on a test trajectory, and a log likelihood score is returned. This mode can be used to evaluate the trained model on untrained test data.

### Selection of k values for the HMM comparison of fast and slow virtual NIR areas

In the comparison of segmentation on the fast and slow virtual areas, we based values on those determined for different cortical areas by [8]. They determined that for an fMRI signal of 1976 TRs, early visual segmented optimally with 119 events, thus 16.6 TRs per event. PMC was optimal with 44 events, thus 44.9 TRs per event. The Not the Fall fMRI signal has a duration of 365 TRs. This gives us 365TRs/16.6TRs per event: k = 22 for the fast area, and 365TRs/44.9TRs per event: k = 8 for the slow area. We use these values in segmenting the NIR activity based on the Not the Fall transcript which has 682 TRs.

In order to make a more systematic evaluation of k values for the fast and slow area we performed an exhaustive analysis with an ensemble of k values varying from 2 to 40 in a grid search, to evaluate the segmentation effect as specified in Eq (2) above.

We used the log likelihood of the model fit to evaluate segmentation with different k values. We gathered the log-likelihoods for the fast and slow NIR model, for 50 NIR models, and then exhaustively evaluated the above inequality for all values of k with paired t-test.

### Model code and data

This research is realized in the open code spirit, and indeed benefitted from open code and data for the fMRI experiments [9,44] and the HMM segmentation model [8], and for development of the reservoir model [46], and the language model for word embeddings [39,40]. The Narrative Integration Model code in python, and all required data is available on GitHub https://github.com/pfdominey/Narrative-Integration-Reservoir/.

The fMRI data for 16 subjects in the comparison of human and Narrative Integration Reservoir event segmentation originates from the study of [45]. Data from the angular gyrus for the first 24 minutes of the Sherlock episode were derived from this data and provided by Baldassano https://figshare.com/articles/dataset/Sherlock_data_for_OHBM/12436955. The transcript of the 18 minute auditory recall of the Sherlock episode segment, and fMRI data for the Not the Fall narrative, and the corresponding transcript are described in [44] and are provided in this repository http://datasets.datalad.org/?dir=/labs/hasson/narratives/stimuli.

### Acknowledgments

This research benefited from open access to fMRI data and narrative transcripts [8,44,45,47], the HMM segmentation model [8], and a framework for reservoir computing modeling [46], and word embeddings [39], without which the current work would not have been possible.

## Author Contributions

**Conceptualization:** Peter Ford Dominey.

**Data curation:** Peter Ford Dominey.

**Investigation:** Peter Ford Dominey.

**Methodology:** Peter Ford Dominey.

**Software:** Peter Ford Dominey.

**Validation:** Peter Ford Dominey.

**Visualization:** Peter Ford Dominey.

**Writing – original draft:** Peter Ford Dominey.

**Writing – review & editing:** Peter Ford Dominey.

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
