## [Decision Letter · Decision Letter 0]

17 May 2021

Dear Dr Dominey,

Thank you very much for submitting your manuscript "Narrative Event Segmentation in the Cortical Reservoir" for consideration at PLOS Computational Biology.

As with all papers reviewed by the journal, your manuscript was reviewed by members of the editorial board and by several independent reviewers. In light of the reviews (below this email), we would like to invite the resubmission of a significantly-revised version that takes into account the reviewers' comments.

As you will see both reviewers agreed that your use of the reservoir computing to model (and explain) event segmentation and its hierarchical temporal dynamics is an important contribution. Reviewer 2 also applauded your efforts in the use of the data and model from the Baldassano paper facilitating the comparison between the data and the model. As you will see both reviewers also raise weaknesses that need to be addressed. Some of these are relatively minor, as they relate to omissions of important details or clarity in the presentation but others are more substantial: e.g. statistical assessment (choice of statistical tests) and, related, choice of model parameters (in particular K in HMM). The choice of your model parameters requires a more quantitative and statistical approach (reviewer 1) and you could further quantify the benefits of the entire approach by for example comparing reservoir computing with a linear integrator suggested by reviewer 2.

The clarity of the manuscript could also be improved. Some of this might just be done by reordering (see comments from the reviewers).

The introduction of the reservoir computing (in the Introduction) is particularly terse. The paragraph that starts with "It is not surprising that recurrent neural networks ..." is particularly dense.

We cannot make any decision about publication until we have seen the revised manuscript and your response to the reviewers' comments. Your revised manuscript is also likely to be sent to reviewers for further evaluation.

Sincerely,

Frédéric E. Theunissen

Associate Editor

PLOS Computational Biology

Samuel Gershman

Deputy Editor

PLOS Computational Biology

Reviewer's Responses to Questions

**Comments to the Authors:**

Reviewer #1: Dominey uses a recurrent neural network model to capture several temporal dynamics observed in fMRI data during naturalistic narrative comprehension: (1) context construction at varying times scales, (2) rapid context flushing, and (3) slower-evolving events at higher-level cortical areas. The reservoir network model essentially compresses a sequence of prior semantic vectors into a discourse vector that can be deployed immediately during ongoing processing. This is an interesting (and challenging) paper. Most of my comments pertain to the first half of results validating the event segmentations of the reservoir network; namely, I think the statistical assessment of event segmentions can be improved, and we need more details about certain analytic choices (e.g. the number of events). I also think a more thorough explanation of the reservoir network early on would help readers interpret the results (e.g. how is the network trained? how do the different time constants arise?). I include a handful of minor comments and typos at the end.

Major comments:

As someone who’s not very familiar with this kind of network model, I think the manuscript would benefit from a bit more exposition early on about how the reservoir network works. We get some details in the Discussion and the Methods section located at the end of the paper, and at some points the author points to a previous paper (Uchida et al., 2021); but there should be sufficient “standalone” details about the reservoir network in this paper, and these should come early enough in the Introduction/Results that the reader can more easily interpret the findings. For example, you say “the reservoir was trained to generate the discourse vector”—but how exactly? What’s the training objective? Is the reservoir network previously trained on the text corpus somehow? You mention in the Discussion that in the “current modeling effort we do not train the reservoir,” but this comes very late (and is still a bit opaque to me). Another example: How do the temporal dynamics (time constants) of the network arise? Do they emerge out of some sort of training, or are they user-specified parameter settings when wiring the reservoir? (Apologies if these things are obvious and I’m just missing it!)

The choice of applying event segmentation to downsampled whole-brain fMRI data doesn’t seem to have a strong precedent in the related literature. For example, Baldassano et al., 2017, and 2018, use searchlights and/or focus on high-level association areas like angular gyrus, posterior medial cortex, and medial prefrontal cortex. In this case, putative event representations are encoded in finer-grained response patterns within a given region of interest (although not too fine-grained via Chen et al., 2017). It’s not clear to me that these sort of response patterns seen in the literature contribute meaningfully to the coarse, whole-brain response patterns used here, and I’m somewhat surprised the event structure is evident at the whole-brain level (e.g. Fig. 4A). This non-localized approach also seems to be in tension with the idea of a cortical hierarchy with different temporal dynamics. Replicating this analysis with more localized cortical areas implicated in high-level event representation could strengthen the argument; otherwise, better motivation should be provided for using coarse whole-brain data.

There are several places where the choice of the predefined number of events k for the HMM seems arbitrary or insufficiently motivated. For example, why is a HMM with k = 5 used when applying the reservoir network to the Wikipedia-based test narrative generated from four Wikipedia articles? Were multiple values of k assessed (which values?) and k = 5 chosen (based on what criterion?)? Or did you try k = 4, but the result fit poorly? Other examples: when running the HMM on the fMRI data, you specify k = 10; when you examine the fast and slow reservoir subnetworks, you use k = 25 and k = 8—why? If you’re trying multiple values of k here, it should be a systematic comparison and we need to know the criteria for selecting k; for example, I would consider using the t-distance introduced by Geerligs et al., 2021 (There’s also a nice demo here: https://naturalistic-data.org/content/Event_Segmentation.html).

I’m having a hard time understanding how event boundaries are statistically compared here. For example, you report for the reservoir network that the “ground truth and HMM segment boundaries are highly correlated, with the Pearson correlation r = 0.99, p < 0.0001.” What exactly is being correlated here? Are you correlating a time series of zeros with ones where an event boundary is found? In this case, the degrees of freedom is the number of (autocorrelated) time points. I’m not sure this sort of statistical test is adequate and would advocate for a nonparametric randomization-based approach. For example, Baldassano et al., 2017, use a randomization procedure where they shuffle the boundaries (e.g. 1000 times) while preserving the duration of the events (in conjunction with some metric like the t-distance mentioned above).

In line with the previous comment, when comparing the boundaries (at k = 5) found for the NYT and Wikipedia test narratives, you say you “normalized the resulting event boundaries into a common range, and made a pairwise comparison of the segment boundaries for the two texts.” I’m not really sure what this means. You compared the index of the time points on which each of the four boundaries landed? But your t-value has 5 degrees of freedom, suggesting 6 boundaries were compared… including the first and final time point? Again, I think a nonparametric statistical approach for comparing segmentations (e.g. adapted from Baldassano et al., 2017) would make this more convincing.

In Figure 11, you show the pre-reservoir time-point correlation matrix for Wikipedia2Vec embeddings that serve as input to the network. The lack of slow, event-like structure seems obvious here, but it could be useful to treat this as a more formal “control” model. In other words, if you want to show that the reservoir network captures narrative structure above and beyond the word-level embeddings, it might be worthwhile to show that it provides statistically better event segmentations than the pre-reservoir embeddings.

This paper demonstrates that relatively straightforward recurrent dynamics can reproduce several of the temporal dynamics observed in fMRI data during narrative comprehension. However, the modeling work here doesn’t really touch on the actual content of those high-level event or narrative representations. For example, Baldassano and colleagues relate event representations to situation models. Do we have any interpretation of the discourse vectors represented by the reservoir network (other than summarizing the prior semantic vectors)? You touch on this in the Discussion on page 16, but it might deserve an additional sentence or two.

The figures are described in the main text, but I don’t see any figure captions in the copy of the manuscript provided by the journal or on bioRxiv. Standalone figure captions would be helpful.

Minor comments:

For figures with time-point similarity matrices (e.g. Figs. 2, 4), it would be helpful to see color bars so we have an intuition about the scale of correlations(?) observed.

Abstract: not sure you need “awake” in the first line here

Abstract: expand “HMM” acronym

Page 4: “Wikipedia2ec” > “Wikipedia2Vec”

Page 5: “has to with” > “has to do with”

Page 7: “Braniak” > “BrainIAK”

Figure 7: “Alignement” > “alignment”

Page 8: I would use the full story name and credit the storyteller here: “It’s Not the Fall that Gets You” by Andy Christie (https://themoth.org/stories/its-not-the-fall-that-gets-you)

Page 9: You note the 10 events assigned to the fMRI data and say “Likewise for the Narrative Integration Reservoir, 10 instances were created…” This wording implies to me that the number of reservoir instances relates to the number of events—but I think you want to say that the number of instances is matched to the number of subjects (also 10). Also, you say you “summary results of this segmentation”—but how do you summarize across instances?

Page 14: “remain” > “remaining”

Page 15 “reservoir correlations displayed an abrupt response in the forgetting context” due to “immediate responses to input in the reservoir”—this still wasn’t very intuitive to me… why?

Great to see the code on GitHub!

References:

Baldassano, C., Hasson, U., & Norman, K. A. (2018). Representation of real-world event schemas during narrative perception. Journal of Neuroscience, 38(45), 9689-9699. https://doi.org/10.1523/JNEUROSCI.0251-18.2018

Chen, J., Leong, Y. C., Honey, C. J., Yong, C. H., Norman, K. A., & Hasson, U. (2017). Shared memories reveal shared structure in neural activity across individuals. Nature Neuroscience, 20(1), 115-125. https://doi.org/10.1038/nn.4450

Geerligs, L., van Gerven, M., & Güçlü, U. (2021). Detecting neural state transitions underlying event segmentation. NeuroImage, 236, 118085. https://doi.org/10.1016/j.neuroimage.2021.118085

Samuel A. Nastase

Reviewer #2: In this manuscript, Dominey presents a reservoir-computing model (a “narrative integration reservoir”) of event-boundary and temporal-integration processes in the human cerebral cortex. In particular, this model seeks to explain neural phenomena related to “hierarchical event segmentation” (as reported by Baldassano et al., 2017) and “context construction” (as reported by Chien & Honey, 2021). Broadly, these neural phenomena relate to (i) how quickly the state-vectors of cortical regions change at the boundaries between “events” in a narrative [e.g. changes of scene] and (ii) for how long the state-vectors within a segment of the narrative display context-dependence [e.g. where the state-vectors for the current input depend on the state of input 6 words earlier, or whether they are independent of the prior input].

The neural data are modeled in a two-stage procedure. In the first stage, the narrative stimuli used in the original studies is converted to a sequence of vector, via word-by-word mapping using the wikipedia2vec embedding procedure (Yamada et al. , 2020). In the second stage, the embedding sequences are converted into neural state sequences by feeding them through a reservoir model. In this way, the model is able to generate “state transitions” and “context dependence” effects that match the effects from the literature.

The manuscript demonstrates that the reservoir model generates event boundaries comparable to those detected by the HMM model in the original fMRI data, and the unit activation and pattern correlation seem to follow the gradual context construction and rapid context forgetting. Finally, the units within the reservoir model were able to be grouped according to different timescales, enabling the model to replicate the hierarchical pattern of event segmentation, context construction and context forgetting.

The main strengths of this manuscript are that:

i) the research topic and the modeling approach are important: event segmentation and multi-timescale dynamics are highly active areas with broad interest to researchers studying human computational and cognitive neurosciences

ii) the reservoir model is an intriguing and exciting model of temporal integration in the cerebral cortex, because it does not rely on precise tuning of the model parameters, and instead achieves its power via the high-dimensionality of its own dynamics; the tolerance for variation in the weights and in the dynamics makes this computational approach biologically plausible, and it could even help to account for some of the redundancy and plasticity that is observed in cortical-dependent function;

iii) the paper directly employs many of the same techniques from the original papers it is modeling, nicely clarifying the comparison between model and data;

iv) the paper is written in a lively and compelling style.

The main weaknesses of this manuscript are that:

i) the paper does not attempt to isolate the key functional features that account for the effects — in other words, the manuscript does not describe how the reservoir model be adjusted so that it does /not/ work. In particular, a comparison with a linear integrator model (rather than a nonlinear reservoir model) would be highly instructive;

ii) the manuscript claims that there is no learning / training involved, but the use of wikipedia2vec surely involves a rich form of learning; more generally, more discussion is needed of whether the authors are claiming that the event-grouping effects in fMRI data are simply reflections of the within-event semantic similarity, or whether the event transitions may reflect other factors;

iii) related to the point above, the manuscript does not sufficiently characterize the autocorrelation that is present in the embedding vectors themselves… for example, the correlation between words within a wikipedia article must surely be higher than the correlation between words across distinct articles; it is possible that the reservoir model may magnify these effects, but a precise characterization of the autocorrelation before and after transformation through the reservoir model is critical;

iv) there are some smaller technical issues about how the model results are compared with the empirical data, as outlined below.

Overall, this is an intriguing manuscript with great potential, following revisions, to advance our understanding of event-segmentation and temporal integration processes in the human brain.

———

MAIN POINTS

1) The most important missing piece in this manuscript is to present a comparison model that does /not/ produce the effects exhibited by the narrative reservoir model. In particular, it is crucial to demonstrate that the model’s performance is different from something very simple like feeing the embeddings into a linear integrator model, or a linear filter [e.g. a boxcar running average, or a recency-weighted average like [0.2, 0.18, 0.16, 0.14, 0.12, 0.08, 0.06, 0.04, 0.02] . More generally, it would be fascinating to know which of the elements of the reservoir model are necessary in order to account for the effect: for example, what happens if leak rate is set to near-zero, or if it is set to a very high value? Together such comparisons could help to understand whether nonlinearity is even required to generate the effects, and to separate out which effects arise from the reservoir component, and which effects arise from the wikipedia2vec embedding model.

2) Related to the point above, the manuscript claims that the results do not require any learning. For example, in this paragraph: “It is worth noting, in the context of the relation between narrative structure and recurrent network functional neuroanatomy, that the observed reservoir behavior comes directly from the input-driven dynamics of the reservoir. There is no learning, and we do not look at the readout, but at the recurrent elements of the reservoir itself.”

Although I understand that the reservoir model is not trained in any way to match the neural data, the model does have the benefit of the enormous amount of information (learned from text and semantic-structure) in the encoding model. At a conceptual level, please textually clarify the sense in which there is no learning. At a more practical level, it seems critical to better characterize how much event-structure can be directly extracted from the embeddings themselves. Figure 11 shows a comparison of the word-embedding autocorrelation structure and the reservoir-state autocorrelation structure, and these do appear to be very different. However, they are plotted on different scales, and there is no smoothing at all applied to the word-embeddings. It seems implausible that words sampled from within one Wikipedia article (or New York Time article) are going to have the same average inter-word similarity as words samples across two distinct articles — certainly many words will be shared or non-specific, but in the time-averaged data, there should be some semantic “themes” that are shared across sentences within an article, but not across articles. It is crucial to separate out such effects [inherent in the input to the narrative reservoir model] from the effects that arise from the recurrent dynamics of the reservoir model.

3) The HMM segmentation model will probably have a bias to “equally space” its events across an interval from start to end — in other words, in the absence of any actual structure over time, such as in random noise data, the HMM will likely segment a sequence into units of similar length. Therefore, in order to compare segmentations in the neural data and in the model, it seems important to run a control in which we use the HMM to cluster a /permutation/ of the real data, and then show that the HMM fit on this permuted data has a lower correspondence to our model [or fMRI data] than the HMM fit on the original. For example, if we have a sequence of events ABCD in the simulation and the same sequence ABCD in the neural data, then we cannot just show that the timepoints of the segmentations are correlated — the more compelling demonstration would be [for example] to compare (i) correlation when simulation and neural data are both using ABCD ordering and (ii) correlation when the simulation uses DBAC ordering and the neural data is for ABCD ordering.

4) There are some issues relating the timing of the words in the (auditory, spoken stimulus, for which the neural data were recorded) and the location of words within the text-transcript fed into the word embedding model. Words are not spoken at a constant rate in a narrative, to 50% of the words do not correspond precisely to 50% of the time in a narrative. In order to align neural data (fMRI timing) with model predictions (word-embedding timing), the only solution is to determine when each word is spoken. The authors propose two methods for aligning the neural data with the stimulus timing, but (as far as I can tell?) neither of these methods actually precisely aligns the timing of the neural data with the timing of when the actual words were spoken. Since the stimuli are all available, is it not possible to generate the simulations in a way that matches directly with when the words were spoken (and when the brain responses were recorded)?

5) When illustrating the time-time correlation maps, it may be helpful to do so only after subtracting the mean value from each element (i.e., for each unit in the reservoir, or each dimension in the word-embedding, treat the signal as a time-series, and subtract the mean of the time-series from every time-point). This can prevent saturation issues which may confuse the interpretation. For example, in Figure 11 there is an illustration of the time-time correlation for the embedding inputs (panel B) and for the reservoir states (panel D). One correlation map is shown on a scale from 0.0 to 1.0 and the other maps is shown on a scale from 0.88 to 1.0. If the mean values are removed from all elements, then the maps will not have these saturation effects, and they can plotted on comparable scales. The saturation effect (where all correlation values are very high) arises when there is a common “mean signal” that is stable within a system over time, so that all pairs of timepoints are highly correlated.

6) The forgetting curves, grouped by timescales, shown in Figure 9, exhibit a dip at the beginning and then a spike around t =15. It is not entirely clear what is happening at t=0 and what is happening at t=15. Please could you make this clearer, both in the text on the figure using labels. Given that the model is driven by word embedding, is it possible that sudden “separation” or “convergence” effects arise from high/low frequency characters (e.g., special characters such as punctuation, or onset-related words) that occur at the event boundary? It is important to rule out the possibility that the model is driven to an unusual state by distinctive words or characters that occur near event boundaries, rather than by the broader “meaning incompatibility” between the prior context and the current input. Does this sudden dip (Figure 9) occur for all event boundaries and/or for all sentence boundaries? [Of course, it does seem that the patterns of the units were driven by the stimulus after the event boundary, and the curves seem to separate over time indicating their representation are gradually different.] 

7) It was unclear to me whether this manuscript is proposing that reservoir dynamics are actually proposed as a process model for cortical dynamics in the human brain. If so, please clarify what are the architectural features that are being proposed -- what are the essential features of the reservoir that we should interpret as functional principles for the cerebral cortex?

8) I was intrigued by this section in the manuscript: “ We can propose that this segregation is a natural product of neuroanatomy, and that narrative structure has found its way into this structure. Narrative structure reflects the neuroanatomy.” Please could you extend and/or clarify this statement. As I understand it, the claim is that (i) each different stage of cortical processing could employ its own reservoir network and (ii) time constants in the reservoirs would be longer in higher order regions, and then (iii) this configuration would explain the results of Baldassano et al who analyzed narrative structure at multiple scales. If this is indeed the logic, please could you unpack this for the reader.

MINOR POINTS

In relation to the final “generalization” analysis, examining the long-vs-short scale event segmentation generalization, please provide a little more information about the generalization accuracy (beyond the difference between coherent vs. incoherent condition). For example, what is the generalization accuracy of event segmentation in each condition? Is there an accuracy difference between long vs. short timescale “regions”? Were meaningful events being segmented in the short- vs. long-timescale “regions”? Providing this information could help validate this analysis and clarify its meaning. Relatedly, the logic and procedure for the HMM-generalization analysis could be clarified in the text. The hypothesis being tested is described as follows: ““We can now test the hypothesis that there is a relation between the time constant for construction, and the granularity of segmentation in the Narrative Integration Model by the HMM.” I think it may be clearer if you phrase this without reference to the HMM, since the HMM is just a data-analysis tool [it is not a process model for brain dynamics] — so I think that the underlying claim here is that differences in timescales of different units (within a reservoir model) can explain the increasing-scale event segmentation effects reported by Baldassano et al?

Figures: Please add axis labels to all panels of all figures, and ensure that the resolution is high; some figures are difficult to read (at least in the format supplied to reviewers)

In Figure 4B, what is the low inter-event correlation (blue cross-shape) that happens in the middle of the story? Is there something very different happening over this period? 

Figure 5 nicely shows the effect of context forgetting and construction on the model representation. However, I find it hard to tell whether different units really “forget” the context at a similar rate. The forgetting curves and pattern correlation seem to take ~20 TRs to reach different activation/low pattern correlation. Is it possible to make this differentiation clearer or more explicit? Furthermore, although the color scheme allows to differentiate the correlation values, all the values are very high (above 0.95). This makes it harder to compare with the neuroimaging results in Chien and Honey (2020) where they subtracted from each voxel its mean signal.

Figures 5 and 6: The figure titles, captions and body text could be clearer on providing a summary “take home message” of these Figures.

Figure 6: In order to demonstrate that construction time and forgetting time are not related, would it not be more direct to plot forgetting-time vs construction-time unit-by-unit?

Figure 6: there is no legend label for panels C and D

Figures 8 and 9: I appreciated the finding that units could be grouped into different timescales based on context construction analysis. Given that the mapped timescale only ranges from 0-35, consider using just 3 groups to make the inter-group difference clearer. I did not perceive much difference between the 5 correlation maps shown in Figure 8, for example.

Figures 9-13: y-axis labels should state “activation difference” or similar (instead of “activation”) 

For the HMM hyperparameters, why was K = 5 chosen for Wikipedia, which also has 4 events as in the NY TImes text? 

Please elaborate on the initialization parameters for the narrative integration reservoir. What is the distribution of values from which W_in and W are randomly initialized? Does this choice powerfully shape the behavior of the reservoir, or would similar results be observed with any other choice, as long as the reservoir dynamics are neither diverging nor collapsing?

**Have the authors made all data and (if applicable) computational code underlying the findings in their manuscript fully available?**

Reviewer #1: Yes

Reviewer #2: None

PLOS authors have the option to publish the peer review history of their article (what does this mean?). If published, this will include your full peer review and any attached files.

Reviewer #1: **Yes: **Samuel A. Nastase

Reviewer #2: No
---

## [Decision Letter · Decision Letter 1]

1 Aug 2021

Dear Dr Dominey,

Thank you very much for submitting your manuscript "Narrative Event Segmentation in the Cortical Reservoir" for consideration at PLOS Computational Biology. As with all papers reviewed by the journal, your manuscript was reviewed by members of the editorial board and by several independent reviewers. The reviewers appreciated the attention to an important topic. Based on the reviews, we are likely to accept this manuscript for publication, providing that you modify the manuscript according to the review recommendations.

Please look at the comments of Reviewer #2 - they are quite detailed and relevant. I will look carefully at your reply.

Sincerely,

Frédéric E. Theunissen

Associate Editor

PLOS Computational Biology

Samuel Gershman

Deputy Editor

PLOS Computational Biology

[LINK]

Reviewer's Responses to Questions

**Comments to the Authors:**

Reviewer #1: The author has substantially reworked and improved the manuscript. The Introduction now contains a thorough description of the reservoir architecture used in this manuscript, which clarifies my confusion about what components of the network are fixed vs. learned. Rather than using whole-brain data, the author now uses response patterns from angular gyrus. Some of the seemingly arbitrary analysis choices (e.g. the number of events k) are now better described and motivated, and the statistical treatment is improved by using a randomization-based nonparametric statistical test for evaluating the event segmentations. Importantly, the author now better evaluates the pre-reservoir embeddings and a linear integrator model, both of which serve as baseline (or control) models. I have a couple minor comments:

In comparing the reservoir model to fMRI data, the author switched to a different story (the widely-used Sherlock movie and recall dataset). I’m a little curious why—but I assume it was just a matter of convenience.

I’m curious about the temporal structure inherent in the reservoir network vs. the temporal structure inherent in the stimulus. For example, it could be interesting to scramble the word order of the stimulus prior to supplying it to the reservoir network (following Lerner et al., 2011).

Author summary: “brains are lead through” > “brains are led through”

Page 6: the sentence structure “and for developing the reservoir model” seems awkward

Page 10: “with p values ranging from p = 1.87e-10 for linear integrator 1 to p = 3.22e-02 for NIR 2”—it might be worth unpacking these results a bit more (or comparing models)

Page 10: “This means that the HMM can be used to measure similarity in event structure in neural activity from different stimulus modalities”—what exactly does this mean? We can run HMMs on two related datasets and then examine the event “templates” (i.e. centroids) found by the model across two datasets (but I don’t think that’s done here). Or we can compare the event boundaries, but this requires somehow matching the timing across datasets (e.g. in Chen et al., 2017, they use averaging to match events across perception and recall).

References:

Lerner, Y., Honey, C. J., Silbert, L. J., & Hasson, U. (2011). Topographic mapping of a hierarchy of temporal receptive windows using a narrated story. Journal of Neuroscience, 31(8), 2906-2915. https://doi.org/10.1523/JNEUROSCI.3684-10.2011

Reviewer #2: Thanks to the author for a raft of new analyses and figures, and for actively engaging with all the points raised in the revision.

My only outstanding concerns relate to the Construction/Forgetting analyses, and to some conceptual and methodological points around that.

(Point 1) I think that the text could be a little more precise about what was shown empirically in relation to construction and forgetting, versus the underlying mechanisms. In Chien & Honey (2020), the finding was not precisely that construction was slow while forgetting was fast. Rather, the finding was that the alignment times (“time for construction of shared context”) increased from peripheral regions toward higher-order regions, while separation times (“forgetting”) did not increase in this systematic manner. It is true that, in the HAT model presented in that paper, there is an “event reset” that leads to rapid forgetting of prior context (leading to asymmetric behavior within the same model for construction and forgetting). However, the empirical (fMRI) finding from that study was not as definitive: the finding was just that the forgetting and construction patterns did not covary with each other: slow-constructing regions were not necessarily slow-forgetting regions. This absence of covariation observed in the empirical data is a kind of asymmetry, but it is not quite the same as saying that, for a single model of a single population, that the circuit can “construct slowly” and “forget quickly”. I think that the temporal asymmetry in individual models of individual regions is still an important point worth investigating — and there are plenty of other theoretical reasons to believe that some kind of context-resetting is occurring — but it is also important to be clear that the empirical finding (in relation to constructing-vs-forgetting timescales) was the absence of covariation across regions.

(Point 2) I am having difficulty understanding how the linear integrator is showing a gradual ramping (for Different —> Same) and yet a rapid decrease (for Same —> Different). I am referring here to the top panel of Figure 5, and the following text in the revised manuscript: “Interestingly the same effects of abrupt forgetting and more progressive construction are observed for the linear integrator. This indicates that this asymmetry in constructing and forgetting is a property of leaky integrators systems.” In the “Integrator Details” section below I elaborate on why I think that linear integrators that are “slow to construct” should also be “slow to forget”; but similar arguments are presented in the Supplemental Material of Chien & Honey (2020).

I am not sure where the discrepancy arises between the linear integrator used in this manuscript and the one used by Chien & Honey (2020). One possibility is that the integrator used here employs a different equation, although this is somewhat ambiguous (see Point 3 below). A second possibility is that the construction/forgetting effects interact in a crucial way with the autocorrelation in the input signals — the simulations in the current manuscript employ correlated input from the word embeddings, while the theoretical arguments of Chien & Honey (2020) made use of uncorrelated input signals. To test for the role of autocorrelated input, one possibility would be to re-run the construction / forgetting analysis on “fake stimuli” constructed as random and independent sequences of vectors generated by sample Gaussian-distributed random numbers. If the asymmetry goes away, then it seems that the sequential properties of the stimulus play a role. A third possibility is that the dependent variable [“difference in activation level”] used in the Figure 5 analyses is different in some important way from the dependent variable (pattern correlation) used in Chien & Honey (2020). However, this possibility seems unlikely, as the current manuscript also shows similar asymmetry effects using a pattern correlation measure.

(Point 3) The equations describing the leaky integrator seems to contain a typo, so it is not totally clear what form of integration was employed. In the text, the equation reads as:

LI(n+1) = ( (1-alpha)LI(n-1)n + (1+alpha)*embedding(n) )/(n+1)

where I believe there may be an erroneous multiplication of LI by n?

(Point 4) Assuming that a key conceptual contribution of the present manuscript is that the reservoir computing model has a repertoire of multiple time constants, then — to be clear — is the neural reservoir being proposed as a model of multiple stages of cortical processing (each with their own timescale)? This is important because one could also imagine the reservoir as a model of a single region, in which distinct nodes within that region each have their own timescale (as in the work of Bernacchia et al, for example). Please clarify in the text.

~~~~

Integrator Details

~~~~

Can linear integrators be “slow constructors” and “rapid forgetters”?

Informally, I want to say something like “integrators that are slow to forget prior context, should also be slow to absorb new input”.

Somewhat more formally, for the linear integrator, let’s call it L(t) and assume it has a form L(t) = p*L(t-1) + q*S(t), where S(t) is the input Signal. (In the case of this manuscript, the input signal values are the outputs from the embedding model.)

Clearly, once p and q are fixed, L(t) depends only on L(0) and the values of S(t).

Moreover, if we are currently at time t, we can choose a time t-tau from the past, and we can see that L(t) depends both on the values of S(t) before and after that moment. In other words, L(t) depends on S(t) for t < (t - tau) and it also depends on the values of S(t) for t >= (t-tau). We can begin to ask whether the S(t) value before t-tau are more important / influential, altogether, than the values of S(t) after t-tau.

I want to claim that:

if I fix L(0), and treat the inputs S(t) as independent random samples from a given distribution,

then the expected proportion of variance of L(t) that depends on [S(0), S(1), … S(t-tau)] is just a function of tau.

In other words, we could say that V_before(tau) = the proportion of variance of L(t) determined by [S(0), S(1), … S(t-tau)].

And similarly, we could say that V_after(tau) = the proportion of variance of L(t) determined by [S(t-tau+1), S(t-tau+2), … S(t)].

Importantly, V_before(tau) + V_after(tau) = 1, because all of the variance in the L(t) values is determined by the S(t) values.

At one extreme, when tau = 0, then we are considering the effect of all of the input on L(t), and obviously we can explain all of the variance in L(t) if we know all the input, so V_before(0) = 1.

At the other extreme, when tau = t, then we are considering none of the input, and we don’t explain any of the variance in L(t), so V_before(t) = 0.

And as tau increases from 0 towards t, we are looking at input further and further in the past, and we are influencing less and less of the input, so V_before(tau) gradually decreases from 1 down to 0, while V_after(tau) gradually increases from 0 to 1.

Because this function V_before(tau) (which is equivalent to 1- V_after(tau)) is just a property of the linear integrator, it will act in the same way regardless of whether we are “entering” or “exiting” a shared context. Therefore, a linear integrator should not be able to exhibit immediate forgetting (rapid) and progressive constructing (slow), as shown in Figure 5.

To make this a bit more concrete, suppose that the Intact models receive the following input sequence:

[A B][C D][E F]

where each capital letter represents some sequence of words in the input stream, S(t), and each capital letter stands in for an equal number of words, and the square brackets indicate the “segments” that were scrambled in the experimental manipulation of the input data.

In that case, we can imagine that the Scrambled group sees the following sequence:

[W X][C D][Y Z]

where the shared portion of the text is “CD” and the preceding and following segments are different from what is seen by the Intact group.

The relevant time period for the “Construction” measurement is the Different-to-Same period leading into the shared segment “CD”: this is BC for the Intact condition and XC for the Scrambled condition.

Similarly, the relevant time period for the “Forgetting” measurement is Same-to-Different period as they exit the shared segment “CD”: this is DE for the Intact condition and DY for the the Scrambled condition.

So when we conduct the Construction analysis, we are asking how much of the variance in L(t) depends on the shared current input (C), even though the preceding contexts (B and X) are different. Thus, for a linear integrator, the construction analysis is a way of measuring V_after(tau).

And when we conduct the Forgetting analysis, we are asking how much of the variance in L(t) depends on the shared history (D), even though the current input streams (E and Y) are different. Thus, for a linear integrator, the forgetting analysis is a way of measuring V_before(tau).

I hope it is clear that, for both the forgetting and construction analyses, the difference (on average) of the states of the Intact and Scrambled models is just a function of V_before(tau) and V_after(tau). But since V_before(tau) + V_after(tau) = 1, we are really measuring the same thing in both cases. If V_before(tau) ramps downward quickly as a function of tau, then V_after(tau) ramps upward quickly as a function of tau.

The reasoning above is by no means a formal proof, but I hope it makes clear why I was surprised by the linear integrator results in Figure 5, and if I am going wrong here, I hope that the author can help me to understand where.

In either case, thanks for the science, and apologies for this long explanation!

Chris Honey

**Have the authors made all data and (if applicable) computational code underlying the findings in their manuscript fully available?**

Reviewer #1: Yes

Reviewer #2: Yes

PLOS authors have the option to publish the peer review history of their article (what does this mean?). If published, this will include your full peer review and any attached files.

Reviewer #1: **Yes: **Samuel A. Nastase

Reviewer #2: No

Figure Files:

Data Requirements:

Reproducibility:

References:

---

## [Editor Report · Decision Letter 2]

8 Sep 2021

Dear Dr Dominey,

We are pleased to inform you that your manuscript 'Narrative Event Segmentation in the Cortical Reservoir' has been provisionally accepted for publication in PLOS Computational Biology.

Best regards,

Frédéric E. Theunissen

Associate Editor

PLOS Computational Biology

Samuel Gershman

Deputy Editor

PLOS Computational Biology

Dear Peter,

I have read your second revisions and your reply to the reviews. Congratulations on a nice contribution.

Frederic Theunissen

---

## [Editor Report · Acceptance letter]

22 Sep 2021

PCOMPBIOL-D-21-00645R2 

Narrative Event Segmentation in the Cortical Reservoir

Dear Dr Dominey,

I am pleased to inform you that your manuscript has been formally accepted for publication in PLOS Computational Biology. Your manuscript is now with our production department and you will be notified of the publication date in due course.

With kind regards,

Andrea Szabo
